# Explaining Deep Learning-Based Driver Models

**Maria Paz Sesmero Lorente** *[ID], **Elena Magán Lopez** [ID], **Laura Alvarez Florez**, **Agapito Ledezma Espino** [ID], **José Antonio Iglesias Martínez** [ID] and **Araceli Sanchis de Miguel** [ID]

Computer Science Department, Universidad Carlos III de Madrid, 28911 Madrid, Spain;
emagan@inf.uc3m.es (E.M.L.); 100363965@alumnos.uc3m.es (L.A.F.); ledezma@inf.uc3m.es (A.L.E.);
jiglesia@inf.uc3m.es (J.A.I.M.); masm@inf.uc3m.es (A.S.d.M.)
* Correspondence: msesmero@inf.uc3m.es; Tel.: +34-91-624-91-11

**Abstract:** Different systems based on Artificial Intelligence (AI) techniques are currently used in relevant areas such as healthcare, cybersecurity, natural language processing, and self-driving cars. However, many of these systems are developed with "black box" AI, which makes it difficult to explain how they work. For this reason, explainability and interpretability are key factors that need to be taken into consideration in the development of AI systems in critical areas. In addition, different contexts produce different explainability needs which must be met. Against this background, Explainable Artificial Intelligence (XAI) appears to be able to address and solve this situation. In the field of automated driving, XAI is particularly needed because the level of automation is constantly increasing according to the development of AI techniques. For this reason, the field of XAI in the context of automated driving is of particular interest. In this paper, we propose the use of an explainable intelligence technique in the understanding of some of the tasks involved in the development of advanced driver-assistance systems (ADAS). Since ADAS assist drivers in driving functions, it is essential to know the reason for the decisions taken. In addition, trusted AI is the cornerstone of the confidence needed in this research area. Thus, due to the complexity and the different variables that are part of the decision-making process, this paper focuses on two specific tasks in this area: the detection of emotions and the distractions of drivers. The results obtained are promising and show the capacity of the explainable artificial techniques in the different tasks of the proposed environments.

**Keywords:** Explainable Artificial Intelligence (XAI); advanced driver-assistance system (ADAS); automotive environment; behavior driver recognition; emotions driver recognition; XRAI (Region-based saliency method)

## 1. Introduction

In the last few years, Artificial Intelligence (AI) computational methods, such as neural networks or knowledge-based systems, have been increasingly applied to different fields with generally excellent results. Some of these fields are related to areas such as healthcare, cybersecurity, natural language processing, and self-driving cars. There are different AI paradigms that group several techniques, one of these paradigms divides these techniques into sub-symbolic and symbolic approaches. The AI started representing the world with symbols, so the first AI techniques were related to symbolic methods that were easily interpretable, such as expert systems or rule-based methods. However, the latest techniques brought by sub-symbolism, such as ensembles or Deep Neural Networks, are related to "black box" techniques whose outputs are difficult to explain.

In this sense, the interpretability and explainability of the methods are currently key factors that need to be taken into consideration in the development of AI systems. In an AI domain, interpretability is the ability of an algorithm to present itself in terms understandable to a human [1], and explainability can be defined as giving human-understandable

motivations of how given attributes of an individual are related to its model prediction [2]. Thus, explainability goes a step further than interpretability by finding a human-comprehensive way to understand the decisions made by the algorithm. In this sense, it is remarkable that in order to fully gauge the potential of the AI, systems of trust are needed [3].

In this context, Explainable Artificial Intelligence (XAI) appears to face one of the main barriers of Machine Learning (ML)—a branch of Artificial Intelligence—related to its practical implementation. As is explained in [4], the inability to explain the reasons by which ML algorithms perform as well as they do is a problem with two causes: The first is the difficulty of using the newest ML models in sectors that have traditionally lagged behind in the digital transformation of their processes, such as banking, finances, security, and health, among many others. The second problem is that the results and performance metrics are not sufficient for the current science and society, and the search for understanding is essential for the model's improvement and practical utility.

In the field of automated driving, XAI is of particular interest because the level of automation is constantly increasing according to the development of AI techniques. In this paper, we focus on the context of Advanced Driver Assistance Systems (ADASs), which are electronic systems that are designed to support drivers in their driving task. This support ranges from presenting simple information to advanced assisting that can even take over the tasks of drivers in critical situations. In a previous work [5], an architecture model based on a multi-agent paradigm for the integration and cooperation of environment ADASs and driver monitors was presented. The research proposed in this paper could be included as part of that architecture.

In this background, the idea of explaining how and why these systems assist the driver is useful since it helps us to debug the different models used by the ADAS and validate the decisions they make. However, these systems and the different tasks that are part of the decision-making process are very complex. Figure 1 shows the different modules that make up an ADAS [6]. Our paper is focused on the *Driver Model* module, and, in particular, on two specific tasks needed in the development of an ADAS: the detection of emotions and the distractions of drivers.

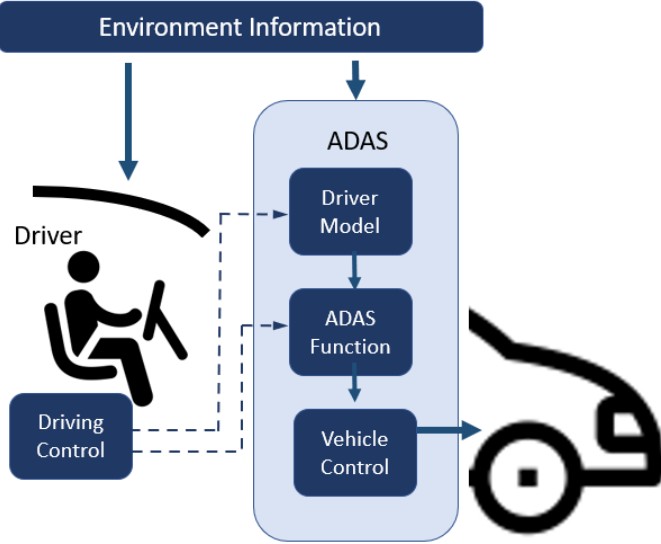

**Figure 1.** Personalized ADAS with a pre-trained driver model.

The two tasks developed in this research have already been faced by using Deep Neural Networks (DNNs), models that map input data to their associated output data through some data transformation in the hidden layers. In general, the results obtained by DNN are relevant and good enough to be used in a real ADAS. However, the nature of the

DNN is a black box, and it makes it difficult for them to be used in critical tasks because of their lack of trust.

In this work, a model of driver emotion detection and a model of driver activity detection [7] will be analyzed using XAI techniques in order to explain how these models work. An explanation of these models, which are an important part of an ADAS, can serve many different purposes, such as debugging them and identifying possible improvements. More precisely, the two different models considered in this research are a model that has been trained to detect the driver's mood and a model that can detect whether the driver is attentive or distracted (for example, texting on the phone). These models receive images of the driver, analyze them by using a Deep Neural Network, and obtain their predictions about the driver's mood and his/her activity. Since Deep Neural Networks are complex black-box models, using XAI techniques, it can be understood why both models are making their decisions, so it will be possible to improve both their accuracy and the trust in the models in future versions of the system. The objective proposed in this research is ambitious, but also very necessary.

This paper is organized as follows: Section 2 provides an overview of the background and related work. The application of an XAI technique in driver modeling is explained in detail in Section 3. Section 4 describes the experimental setting and the experimental results obtained. Finally, Section 5 contains future work and concluding remarks.

## 2. Background and Related Work

In this section, the different research areas related to the background of the paper are described in detail. First, some research work about driver behavior, distractions, and emotion recognition will be described. Afterwards, the most relevant aspects about XAI and its use in the automotive environment are detailed. Finally, the most relevant XAI techniques in the driver behavior modeling are explained.

### 2.1. Driver Behavior and Distraction Recognition

Advanced Driver Assistance Systems (ADASs) are systems that assist the driver in order to increase safety in the car and, more generally, on the road. In Europe, in 1986, different automobile companies and research institutes worked on the Prometheus project (Programme for a European traffic of highest efficiency and unprecedented safety), which proposed solutions, especially to traffic problems [8]. However, at that time, the required technology was not sufficiently mature, and it has only been in the last two decades that this type of research has made important advances. Currently, research has focused on the development of an ADAS capable of intervening in various ways in order to avoid potential danger [8] and thus increasing vehicle safety [9]. In this sense, since 94% of traffic accidents are caused by human error [10], research in all areas related to ADAS development is essential.

Driver-based ADASs are those that incorporate information obtained from the driver to provide any kind of assistance [11]. The incorporation of this type of driver information is crucial; for example, a lane departure detection system that does not integrate driver information cannot detect whether the vehicle departure is intentional or an error. Thus, if the driver is being monitored and drowsiness is detected, it can be deduced that the vehicle departure is unintentional, and some kind of warning could be triggered.

For this reason, behavior driver monitoring is a key aspect of driver assistance. Thus, one of the reasons for the importance of driver modeling is the development of technologies related to driver safety so that they can be incorporated into an ADAS. However, in addition, driver modeling is important because it allows for the emulation of human behavior [12] in the development of autonomous vehicles and because it will allow us to understand how humans behave when driving on the road with autonomous cars [13].

Driver behavior modeling can be classified as reactive or predictive [14]. Reactive models consider the driver's behavior after the action has already been performed. For example, driver training systems employ reactive models to identify danger situations

generated by the driver. On the other hand, predictive models are more complex since they need to identify the driver's actions in real time in order to assist the driver in dangerous situations. A wide variety of models have been proposed and developed to predict, among other aspects, driving maneuvers [15,16], driver actions [17], driver intentions [18], driver states [19], and driver emotions [20].

In relation to the detection of distractions, it is important to take into account that a distraction, unlike inattention, is related to a certain activity (e.g., talking on the phone or turning on the radio) that affects the driver's attention but is not related to the driver's state [13]. In [21], different deep learning-based methods to classify driver's distractions (such as texting, talking on the phone, operating the radio, drinking, and reaching behind) using data from 2D cameras are compared. In [22], the use of data augmentation and Convolutional Neural Networks (CNNs) are shown to be effective at the recognition of distractions and improving classification results, while also reducing training time. With the same purpose, some techniques based on tracking the driver's gaze and attention using head position are proposed in [23]. The result of training a CNN (named HandyNet), which is capable of detecting, segmenting, and localizing (in 3D) driver hands inside a vehicle cabin, is proposed in [24]. In that research, it is used segmenting and tracking hands through the use of depth images and annotation based on chroma-keying. Using more complex setups, in [25], whether prefrontal brain region electroencephalography (EEG) can be used to detect driver's fatigue is examined. According to the authors, although the signal classification accuracy of the prefrontal brain region is not the highest, from a practical perspective, the EEG classification accuracy can be used to detect fatigue. However, in this work, the use of non-invasive sensors is proposed.

Driver Emotion Recognition

Emotions can be defined as states that comprise feelings, physiological changes, expressive behavior, and inclinations to act [26]. As is described in [27], each emotion has unique features related to signal, physiology, and antecedent events. In addition, each emotion also has some characteristics in common with other emotions such as rapid onset, short duration, unbidden occurrence, automatic appraisal, and coherence among responses. The unique characteristics of an emotion are important to differentiate a specific emotion from other affective phenomena. The Facial Action Coding System (FACS) [28] is a comprehensive, anatomically based system for describing all visually discernible facial expressions or emotions [29]. In this sense, in [30], results on the recognition of seven emotional states (neutral, joy, sadness, surprise, anger, fear, and disgust) based on facial expressions are presented.

In recent years, a wide variety of techniques have been used for emotion recognition. In [31], a survey of existing works in emotion recognition using electroencephalography (EEG) signals was proposed. Using also EEG signals, in [32], an approach involving the automatic two-stage classification (negative and positive) and three-stage classification (negative, positive, and neutral) of emotions evoked by music is presented. Moreover, a bimodal emotion recognition system using a combination of facial expressions and speech signals is proposed in [33]. In [34], a deep facial expression recognition algorithm for emotions based on CNNs and an ensemble deep learning algorithm to predict facial expressions are proposed.

In a driving scenario, it is important to analyze the drivers' emotions while driving in order to obtain any kind of information related to their feelings and moods. Psychological studies show that the emotions of the driver play an important role in safe driving [35,36]. In this sense, the authors in [37] proposed that emotions affect driving directly by promoting aggressive driving, and indirectly by reducing the ability to perform several actions at the same time. The authors in [38] proposed a framework for driver emotion recognition using facial expression recognition. The authors in [39] presented an approach for driver emotion recognition involving a set of three physiological signals (electrodermal activity,

skin temperature, and the electrocardiogram). Recently, a complete survey about driver emotion recognition for intelligent vehicles was presented in [40].

This work aims to go further, starting with models based on deep learning that detect both the emotions and the activity of the driver, and using XAI techniques to analyze and explain the decisions made by the system.

### 2.2. Explainable Artificial Intelligence (XAI)

Artificial intelligence (AI) is a very promising field in many different research areas. However, there are several factors that are at play during trust building in AI [41], such as representation, image/perception, reviews from other users, transparency and "explainability", and trialability. In relation to explainability, trusting AI applications is essential to know how these applications have been programmed and how they work in certain conditions. Thus, trust in an AI application is affected if its explainability is poor or missing. Moreover, transparency and explainability are essential not only for building initial trust in AI, but also for continuous trust.

Many of the current AI algorithms exhibit high performance, but they are incomprehensible in terms of explainability. The "black box" is a concept used in machine learning for describing those AI algorithms whose final decision cannot be properly explained. In this sense, there are many areas in which the description of the output of the algorithms needs to be explained in detail to analyze how the algorithms will perform in different situations. For example, the output of an autonomous car offers a clear example of the need for explainable algorithms. The different paradigms behind this framework fall under the umbrella of the so-called explainable artificial intelligence (XAI). The term XAI can be defined as a framework that increases the interpretability of Machine Learning algorithms and their outputs [42]. In addition, XAI is related to those algorithms and techniques that apply AI in a way that the solution can be understood by humans. Thus, the main idea behind XAI systems is that the decisions made or suggested by such systems can be explained with transparency.

The fundamental principles for XAI systems are presented in [43]: explanation, meaningful, explanation accuracy, and knowledge limits. In this sense, AI systems should (1) give reasons for all their outputs (explanation), (2) provide understandable, meaningful explanations that reflect the system's process for generating the output (explanation accuracy), and (3) operate only under conditions for which they were designed.

According to [4], explainability is linked to post-hoc explainability since it covers the techniques used to convert a non-interpretable model into an explainable one. In this sense, XAI is defined in that article as follows: given an audience, an explainable Artificial Intelligence is one that produces details or reasons to make its functioning clear or easy to understand.

### 2.3. XAI and Automotive Environment

The automotive environment is an important field for applying Artificial Intelligence techniques. However, in most of the tasks in which AI is applied in this field, their explainability is essential. For example, autonomous driving is a field of application of AI in which it is necessary to understand the reasons for any mistake made by an autonomous vehicle, and how to fix it. In this regard, XAI is one of a handful of current DARPA programs where machines understand the context and environment in which they operate, and over time build underlying explanatory models that allow them to characterize real world phenomena [44].

Today, Advanced Driver-Assistance Systems (ADAS) are essential in cars since they are becoming increasingly automated. In addition, the amount of responsibility of an ADAS for driving tasks is increasing. In this field, an important aspect to consider is the confidence of the driver in the advanced driver systems. It is essential to provide appropriate explanations for drivers to increase their confidence in the system, which results in appropriate human–AI collaboration [45]. Thus, as is explained in [46], XAI

systems can benefit from having a proper interaction protocol that explains their behaviors to the interacting users. This explanation occurs as a continuous and iterative socio-cognitive process which involves not only a cognitive process but also a social process.

In relation to ADASs, the reliable perception and detection of objects are essential aspects of vehicle autonomy. Deep neural networks are excellent in the detection and classification of objects in images. However, how these networks behave and what the reasons are for their decisions need to be explained in an environment such as automated vehicles. In relation to this aspect, the authors in [47] demonstrated how the results of a deep learning system that detects specific objects for driver assistance in an electric bus can be interpreted. The obtained interpretation explains which parts of the images triggered the decision, and it helps to avoid misdetections. In particular, in that paper, a tool that provides more insight into the performance of a Faster R-CNN framework to understand the reasons for their performance is proposed.

A very important approach in the automotive environment is driver behavior modeling [15]. An important research area in this modeling process is the generation of interpretable models. An interpretable and irrationality-aware human behavior model in interactive driving scenarios is proposed in [48]. The proposed model is based on the cumulative prospect theory (CPT) [49], and the model parameters are learned using a hierarchical learning algorithm based on inverse reinforcement learning [50] and nonlinear logistic regression [51].

In this area, the driver models must be able to justify driver behavior so that passengers, insurance companies, or law enforcement can understand what triggered a particular behavior at an specific moment. Thus, these models need to be explainable and easy to interpret. One possible way to obtain explainable models is from the output given by the driver. Another way would be by analyzing the controller itself (introspective explanations). A related work proposed in [52] tries to generate introspective explanations by detecting the regions of the image that causally influence the network output. In this case, the images are obtained from the environment, not from the driver. In a second step, a video to text model is added that generates explanations of the model's actions: for example, *the car brakes because the road is wet*. To generate these explanations, the authors use the Berkeley DeppDrive-eXplanation database (BDD-X dataset) [53].

*2.4. Exploring XAI Techniques in the Driver Behavior Modeling*

There are several techniques that help us to explain and make models understandable that, by themselves, are difficult for the user to understand. These techniques are called post-hoc explainability techniques, and they contain different approaches: text explanations, visualizations, local explanations, explanations by examples, explanations by simplification, and feature relevance explanations [4].

The aim of this work is to apply techniques that allow humans to understand the decisions made by a system that detects a driver's mood and activity. With this objective in mind, there will be a need for techniques that, first of all, can be applied in image recognition models, specifically in deep neural networks. In addition, instead of simplifying the model or using text explanations, visualization techniques will be useful since they allow us to observe which part of the image is important for decision making. For this reason, we focus on the use of techniques that meet both requirements: visualization techniques compatible with deep neural networks. Some of the most interesting techniques in this context are described as follows:

- LIME [54] is a technique that uses local linear approaches. It provides a specific explanation for each prediction that is accurate locally but does not have to be globally accurate. LIME samples instances close to the one that requires explaining (that is, with small variations in their attributes), and, depending on the class resulting from the sampled instance, a linear approximation that provides a simple explanation is given. In this explanation, each attribute either has a positive weight for a class, in which case it contributes to the classification of that class, or it has a negative weight.

In the case of image recognition, the image is divided into super-pixels, and those that contribute positively to the class are displayed.

- Anchors [55] is a technique that uses local approximations, too. However, in this case, these approximations are not obtained in a linear way, but use if–then rules obtained through a search process. Thus, an *anchor* is an if–then rule, so that, if the conditions established by the *anchor* are met, it is very likely that the prediction will remain stable even if other attributes vary. In this way, *anchors* provide a local explanation for a prediction, revealing the conditions under which the prediction is most likely to be repeated. In the image recognition tasks, it shows the fragment of the image that conditions that prediction (if that fragment appears on a different picture, it is very likely that the predicted class does not vary).

- The SHAP method [56] allows, given an instance, to know which of its attributes have contributed to classifying it, as well as which ones have reduced the possibility of being of that class. To do this, SHAP uses, in combination with a linear approximation, Shapley values [57], which calculate the importance of each attribute of a prediction by making variations of different combinations of attributes and weighting the impact of the modification in the predicted class. In this sense, given an image, it points out the pixels that have been relevant for classification, both for good and bad (as a heatmap).

- DeepLIFT [58] is a specific technique for deep neural networks and analyzes the activations of the neurons in order to discover which attributes are more relevant for the classification of an instance. To do this, DeepLIFT uses a neutral reference against which differences in neuron firings are measured. In this way, the difference between the activation of a neuron for both the reference and the instance is computed, and by using backpropagation, the contribution of the different attributes to the final classification is calculated. In the case of image recognition, the pixels ranked as the most important for a specific prediction can be obtained. As the reference image, it is proposed that several images, such as a distorted version of the original image, are considered to see which is more useful for the model.

- Integrated gradients[59] are a concept similar to the *DeepLIFT* reference image and is called *baseline*, and it is combined with the use of the gradient operation. Thus, to calculate the importance of the attributes, this method computes the gradient of all the points between the *baseline* and the input, and they are accumulated in order to obtain the integrated gradients. In this case, the *baseline* proposed for image recognition is an "empty" image, black in color, but it has the problem that on dark images it can reduce the attribution of dark pixels.

- XRAI [60] is an improvement over the integrated gradients because, although it starts from the same idea (using *baselines*, approximating the importance through integrated gradients), some elements are used that improve its accuracy and understandability. The main improvement is that XRAI divides the image into regions, and the region's importance is then calculated using the integrated gradients. These regions can be of multiple sizes, and they are able to approximate very well the shapes of the images. This way, it is much easier for the users to visualize the parts of the image that have been relevant for the classification, since they can observe the most important sets of pixels, instead of individual pixels with empty spaces between them. In addition, to avoid the black image problem that the integrated gradients had, XRAI uses two *baselines*, one black and one white. The user can also configure the percentage of regions to be displayed (the top 10%, the top 20%...), so that the method can be adjusted to the characteristics of the model. For these reasons, since XRAI is the method that offers the best balance between precision and understandability, we decided to apply XRAI to achieve the goal of this paper.

## 3. Application of XRAI in Driver Modeling

In this section, we describe the target ADAS that is going to be analyzed, and the XAI technique that is going to be used for the explanation.

### 3.1. Target ADAS

As we mentioned before, the objective of this work is to explain an ADAS that keeps track of the driver's mood and activity, and to identify possible improvements. In this section, we explain briefly the two models that belong to this ADAS, since they are extensively described in [7].

Both models are convolutional neural networks, which are a specific subtype of deep neural networks, especially useful for image recognition problems, and were coded using Python, Keras, and Tensorflow. They receive the same image of the driver and analyze it to predict a class: in the case of the "Emotions Model", the class predicted is the mood of the driver, while the "Activity Model" predicts if there is an activity that could be distracting the driver, returning the activity in question.

The complete ADAS in which these models are integrated is designed to be as accessible as possible, so the models will only need a camera placed on the dashboard, in front of the driver, to gather the information they need. By having the camera at this position, the ADAS can see both the driver's environment and the driver's face, so the two models will be able to work with the same input. This camera's position conditions the datasets used to train the models, since they will have to show a frontal view of people.

#### 3.1.1. Emotions Model

The purpose of the emotions model [7] is, given an image of a person's face, predict which is their mood at that moment by analyzing the image. In particular, this model attempts to detect one of seven emotions: happiness, sadness, neutral, fear, anger, surprise, and disgust. To train this model, four different datasets have been used. These datasets have the characteristics specified in Table 1.

**Table 1.** Datasets used to train the emotions model.

| Dataset's Name | No. of Subjects | No. of Samples | Image Size | Specific Information |
|---|---|---|---|---|
| The Extended Cohn-Kanade *Dataset (CK+)* [61,62] | 123 | +10.000 (593 different sequences) | 640 × 490 pixels | Diverse dataset: it includes men and women of different races |
| Facial Expression Recognition *Challenge: FER (2013)* [63] | Undefined | +35.000 | 48 × 48 pixels | It was created by searching in Google the target emotions. A small percentage of the images is mislabelled or does not represent a person |
| The japanese facial expression *(JAFFE)* [64,65] | 10 | 219 | 256 × 256 pixels | The subjects are all Japanese women |
| *The Karolinska Directed Emotional Faces (KDEF)* [66] | 70 | 4900 | 562 × 762 pixels | 50% men, 50% women. Photographs taken from 5 different angles |

However, before these images are analyzed by the convolutional neural network, it is necessary to preprocess them. The preprocessing process applied in [7] is summarized in Figure 2. It should be noted that, for the face recognition step, OpenCV's face recognition library is used.

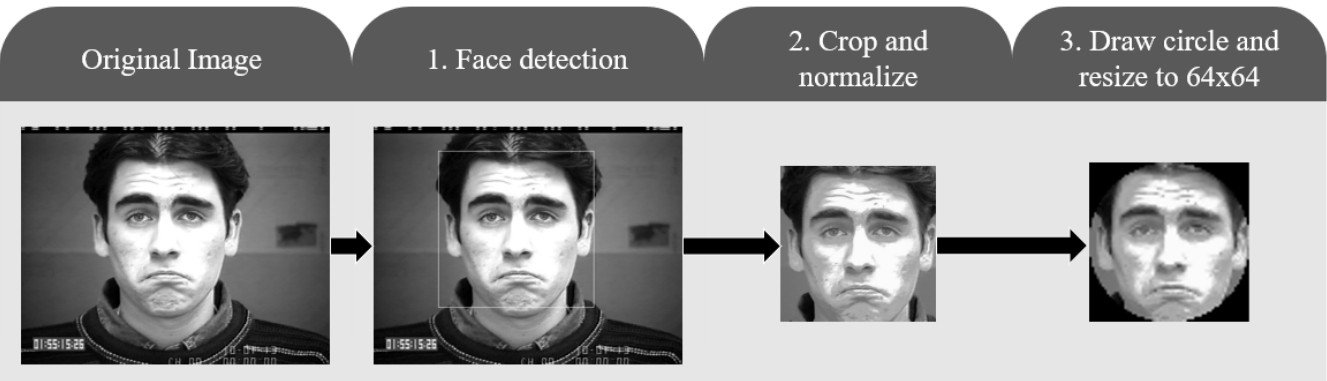

**Figure 2.** Preprocessing example. Photograph taken from the CK+48 dataset (©Jeffrey Cohn).

The preprocessed image, after Step 3, is the input given to the model, which processes it with a convolutional neural network trained with the architecture shown in Figure 3.

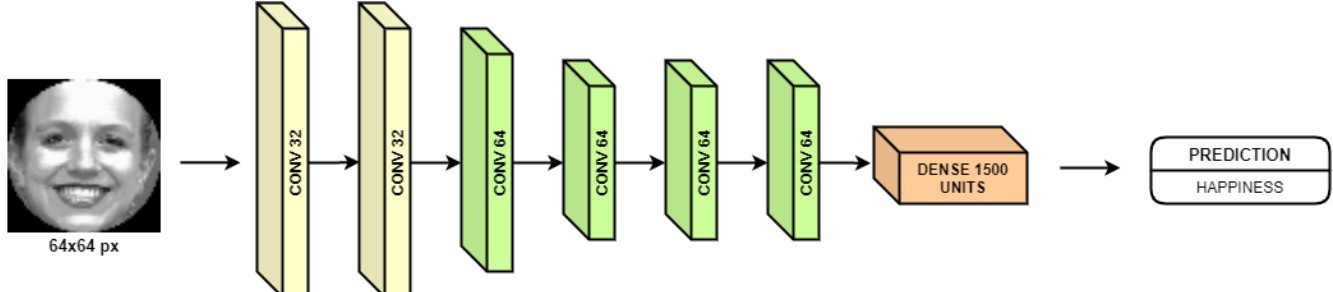

**Figure 3.** 3D representation of emotions model's architecture. Photograph taken from the CK+48 dataset (©Jeffrey Cohn).

### 3.1.2. Activity Model

The purpose of this model [7] is, given an image of a driver, to predict whether the driver is distracted or not. In particular, this model attempts to detect 1 of these 10 common activities: safe driving, visible fatigue, drinking with the right hand, drinking with the left hand, reaching behind, checking GPS, sending a message with the right hand, sending a message with the left hand, talking on the phone with the right hand, and talking on the phone with the left hand.

To train this model, just one dataset was used, since it is the only one that records drivers using a camera located on the dashboard in front of the driver and classifies their activity. Information about this dataset can be found in Table 2.

**Table 2.** Datasets used to train the activity model.

| Dataset's Name | No. of Subjects | No. of Samples | Image Size | Specific Information |
|---|---|---|---|---|
| Multimodal Multiview *and Multispectral Driver Action Dataset (3MDAD)* [67,68] | 50 | +110.000 (16 different activities) | 640 × 480 pixels | Sequences of images that show every frame between the beginning and end of an action. Some frames do not represent the activity they are labelled as. |

This model also requires some preprocessing of the images, but in this case it will be much simpler, since it will only consist of resizing the image to $160 \times 120$ pixels.

In this case, the final architecture of the convolutional neural network can be seen in Figure 4.

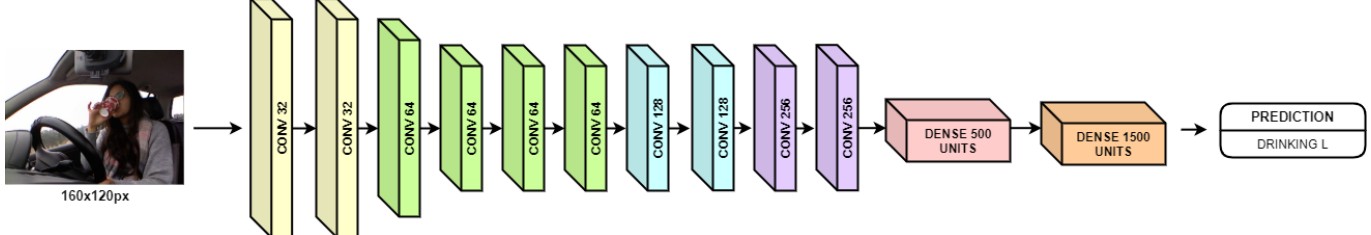

**Figure 4.** 3D representation of the activity model's architecture. Photograph taken from the 3MDAD dataset.

### 3.2. Applied XAI Technique: XRAI

In this subsection, we explain briefly the *XRAI* [60], the XAI technique used to explain the decisions that the models of the ADAS are taking. As we mentioned in the background section, it is a post-hoc explainability technique, which means that it allows one to interpret easily why a black-box model, such as a deep or a convolutional neural network, is making its decisions. It is also a visualization technique, because this method generates explanatory images that aid in comprehending the model. As one would expect from a post-hoc technique, *XRAI* takes the already trained model and adds a new layer of explainability. Because *XRAI* is also implemented using Python and Tensorflow, it is perfectly compatible with the models of the target ADAS.

Introducing *XRAI*'s explainability layer over the model allows us to explain a specific input. If we provide this layer with an image of the driver and the prediction of the driver's mood or activity according to the models, it will point out which part of the image was the most important to determine the models' predictions. To do this, it essentially follows three steps: divide the image into regions, calculate the attributions of these regions, and select which regions are more relevant to the decision.

To divide the input image in regions, *XRAI* first divides the image in six different ways by using Felzenswalb's graph-based method [69] with different scales, so that there are six sets of regions of different sizes. Since these sections' boundaries tend to align with the edges in the image, the regions are dilated so that they include thin edges inside the segment.

After that, the attribution of the regions (that is, their importance for the prediction of that particular input) is calculated. To do that, *XRAI* uses backpropagation in a similar way to the *integrated gradients* technique, except that it uses two baselines (black and white) instead of one (black). To summarize it, it computes the gradient of all the points between the *baselines* and the input, and these gradients are accumulated to obtain the attribution.

Finally, the regions' attributions are evaluated to determine which regions are the most important. This process is relatively simple: the attributions of each region are summed, and those sections that sum to more positive values are the most important. This means that the regions can be sorted by importance, which gives the user the opportunity to choose how many regions they want to see: the top 10% more important, the top 20%, etc.

Once *XRAI* has calculated the regions' importance, we can visualize those sections. One way is representing those regions as a heatmap, which will allow us to notice the importance of the whole image, and the other way is to use the segments as a mask and show only a percentage of the original image based on the importance of each part. Figure 5 shows an example of both visualization methods.

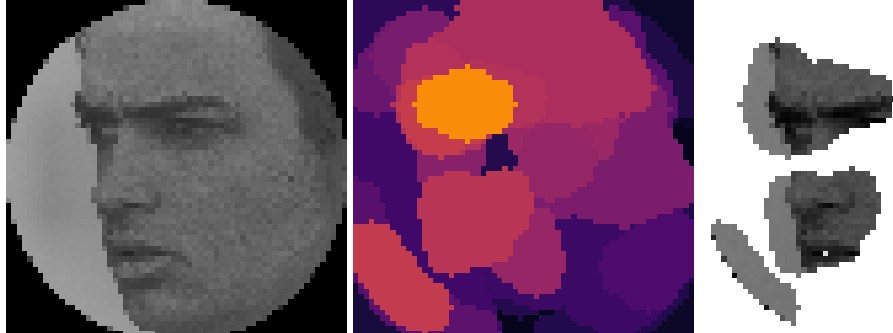

**Figure 5.** (from left to right) Preprocessed input. Heatmap. Top 30% regions. Explanations corresponding to the KDEF image with id AM01ANHL.

To be able to see all the important characteristics of the image, in the Results section, we show the original image along with various heatmaps and segments visible. There are four views of the image where only a small percentage of it, corresponding to the relevant regions, are visible, which are associated with the top 10%, top 15%, top 20%, and top 30% most important regions. This allows us to understand which segments of the image the models are taking into account to make their predictions.

Below the top regions, there are heatmaps with a different color intensity, since details about the regions are more noticeable at one intensity or another depending on the image. Heatmaps show all regions colored on a scale from black to white, where the most important regions appear in a brighter color. The preprocessed image is shown next to both visualizations, in black and white, as a reference.

Figure 6 shows an example of an XRAI visualization.

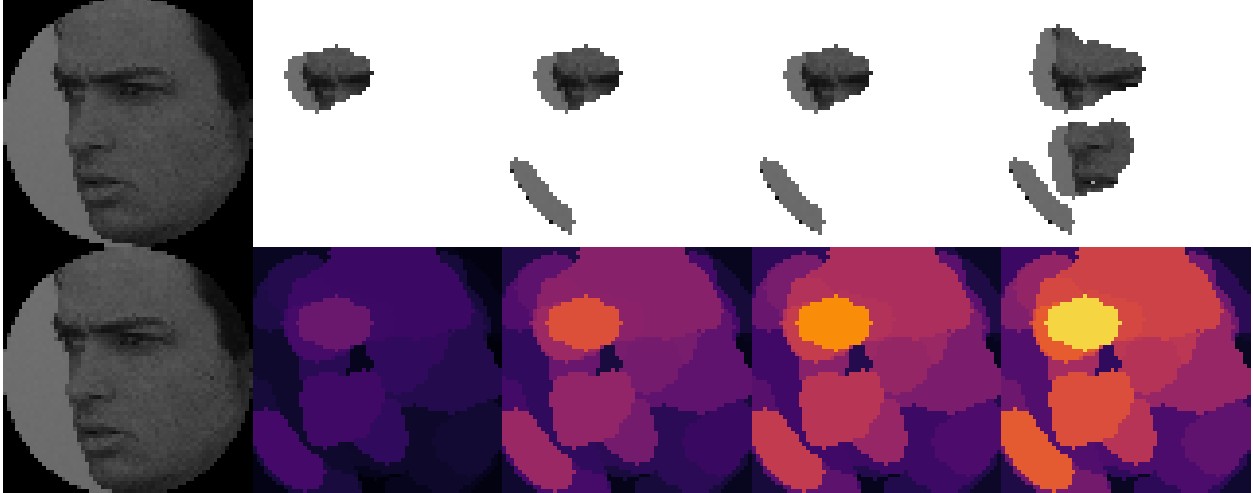

**Figure 6.** Example of an XRAI visualization of an emotion classified as "anger". (up) From left to right: the preprocessed image and the top 10%, top 15%, top 20%, and top 30% most important regions. (down) Heatmaps colored with multiple intensities. Explanations corresponding to the KDEF image with id AM01ANHL.

## 4. Experimental Setup and Results

In this section, we describe the datasets that have been used for testing and explain the results in detail.

### 4.1. Datasets

In this subsection, we present the datasets used to test and debug the model using the *XRAI* technique. There are two kinds of datasets: those that were used to train the models, so that the model classifies correctly, and those that are completely new to the models. We decided to use one dataset of each type for each of the models.

### 4.1.1. Datasets Previously Used for Training

Since these datasets were previously explained in Section 3.1, we only mention which of those datasets will be used for testing:

- *The Karolinska Directed Emotional Faces (KDEF)*: used to test the emotions model;
- *Multimodal Multiview and Multispectral Driver Action Dataset (3MDAD)*: used to test the activity model.

### 4.1.2. New Datasets

It is useful to test the models over data that has not been used for training, since it will prove the ability of the model to generalize and be used in a real, unknown environment. By changing the dataset used for testing, we will force the model to work with images where there are people that it has never seen, and where there can be different illumination conditions, backgrounds, angles, and so on. As with the training datasets, we have chosen two datasets to analyze, one for each model.

To test the emotions model, we decided to use the *Radbound Faces Database (RaFD)* [70]. The Radbound Faces Database consists of photographs taken of adults and children, both male and female, and mostly Caucasian, from five different angles (as *KDEF*) and representing eight different emotions (it adds the "contemptuous" emotional expression, which we will not use in this work). We use images wherein subjects are looking forward, taken at three camera angles: straight, half left profile, and half right profile.

As we mentioned before, to our knowledge, there is only one dataset that records drivers with a camera located in front of them, so there are some limitations to the datasets that we can use. *State Farm Distracted Driver Detection* [71], a dataset from a Kaggle competition, and the *AUC Distracted Driver Dataset* [72,73] are two existing datasets that show images of people doing activities while driving, quite similarly to *3MDAD*. These dataset images show one of 10 possible activities, 7 of which our model should be able to predict: safe driving, drinking, reaching behind, sending a message with the right hand, sending a message with the left hand, talking on the phone with the right hand, and talking on the phone with the left hand. However, there is a main difference with the dataset used for training, which is that these photos have been taken from the driver's side, and not from the front.

Because of this, we tested the activity model with our own images, for which we recreated the angle used in the dataset used for training. We took photographs of seven different subjects performing the distractions considered by the model, for a total of 200 images.

### 4.2. Results and Discussion

In this subsection, we show and discuss the results obtained. To do this, we analyzed the two models (the "emotions model" and the "activity model") separately.

The experimentation was performed using a Python program that could be executed on virtually any computer. This program loads the models, reads a batch of images, processes them, and then provides the XRAI explanation of the predictions. The most relevant aspects of the testing environment are the libraries and packages used, which would be Tensorflow 2.3.1, Keras 2.4.3, Python 3.8.5, and OpenCV 4.4.0.46. We also used the XRAI implementation provided in the Saliency package, version 0.0.5.

Apart from the XRAI explanations, basic accuracies of the models are analyzed along these lines. More detailed results, represented as confusion matrices, can be consulted in Appendix A.

### 4.2.1. Emotions Model's Results

First, we tested the emotions model with the *KDEF* dataset, since the multiple angles of photographs could provide a great amount of information about the model's ability to generalize. However, we limited ourselves to use only three angles—straight, half left profile, and half right profile—since OpenCV's face recognition library did not detect faces

on the full profile pictures. Considering these limitations, we used XRAI on a total of 2938 images.

1666 of these pictures (57%) were incorrectly classified, which is a surprisingly high number if we consider that this dataset was used for training. However, it is important to note that OpenCV's face recognition library did not locate the face in 18% of the samples, and the model had to use the full image as an input instead of the cropped one. This worsens notoriously the model's predictions, since 73% of the images where OpenCV did not provide the face where mislabelled.

There are other failures due to face recognition: for example, in some samples, OpenCV provided face elements of the image, such as an ear or an eye, as observed in Figure 7. This happens mostly in photographs taken from a half profile angle, where this face recognition system seems to perform less accurately, so this will be an element to be improved in future versions of the ADAS.

Among the mislabelled photographs, we find "reasonable confusions", such as the classification of Figure 8, which the model thought represented "HAPPINESS". Although it may seem hard to confuse, after preprocessing the image, the size and quality of the image was reduced, and it was easier for the model to overlook details, such as forehead wrinkles. In Figure 9, we can see that the model sees the mouth of the man and interprets it as a smile.

However, there are other cases where neither the prediction nor the explanation is logical. An example of this can be seen in Figure 10, which shows the explanation of an image labelled as "SAD" that the model classified as "HAPPINESS". As we can see, the model focuses mostly on her nose, nasolabial folds, and cheekbones, which are not representative of the emotion, and so the model fails its prediction.

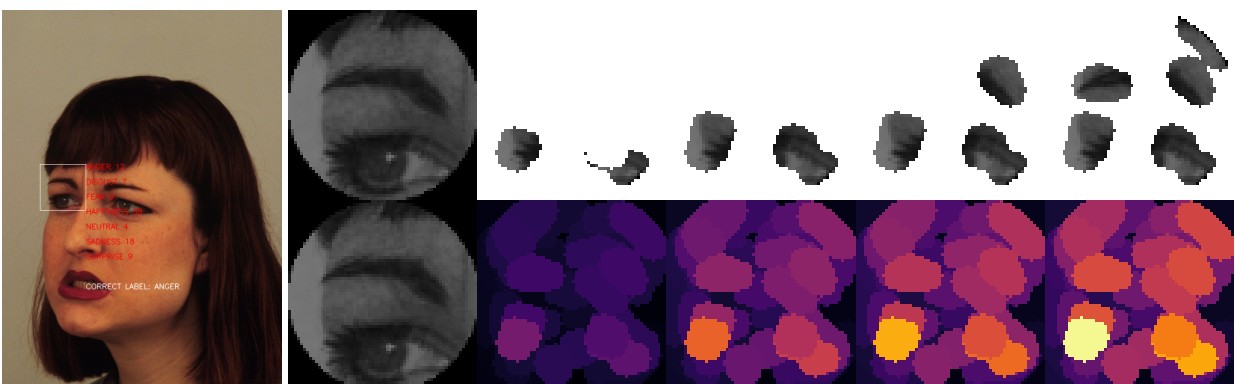

**Figure 7.** Example of face recognition failure. Prediction and explanations corresponding to the KDEF image with id AF10ANHL.

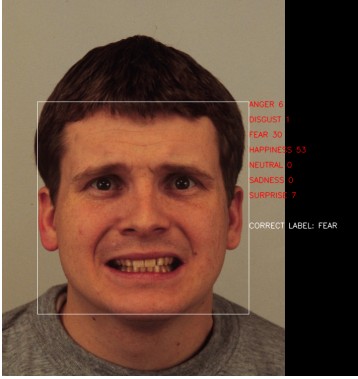

**Figure 8.** Image classified as "HAPPINESS" by the model. Prediction corresponding to the KDEF image with id BM31AFS.

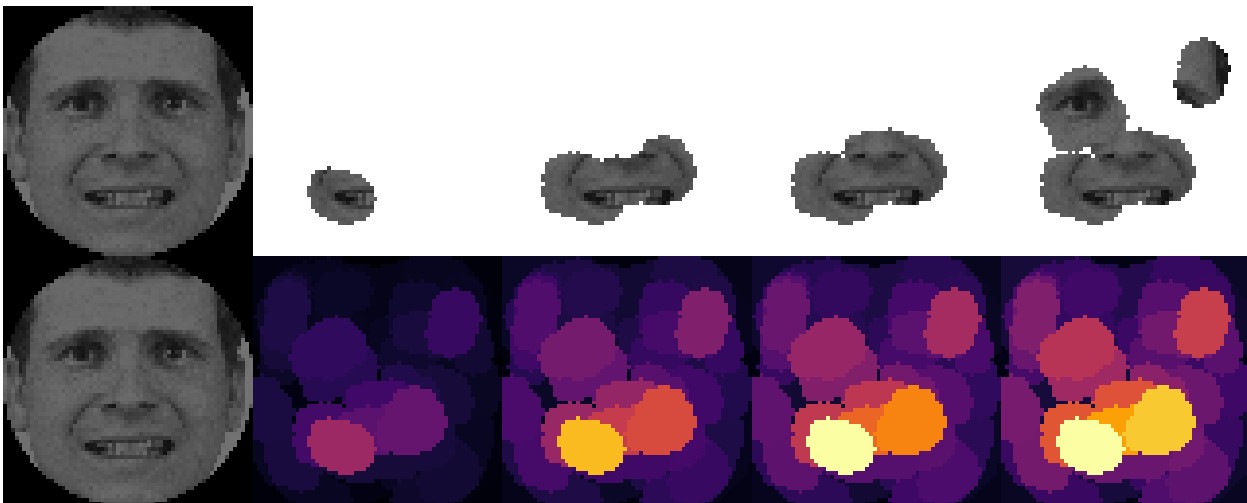

**Figure 9.** XRAI visualization of a man performing emotion "FEAR" that was classified as "HAPPINESS". Explanations corresponding to the KDEF image with id BM31AFS.

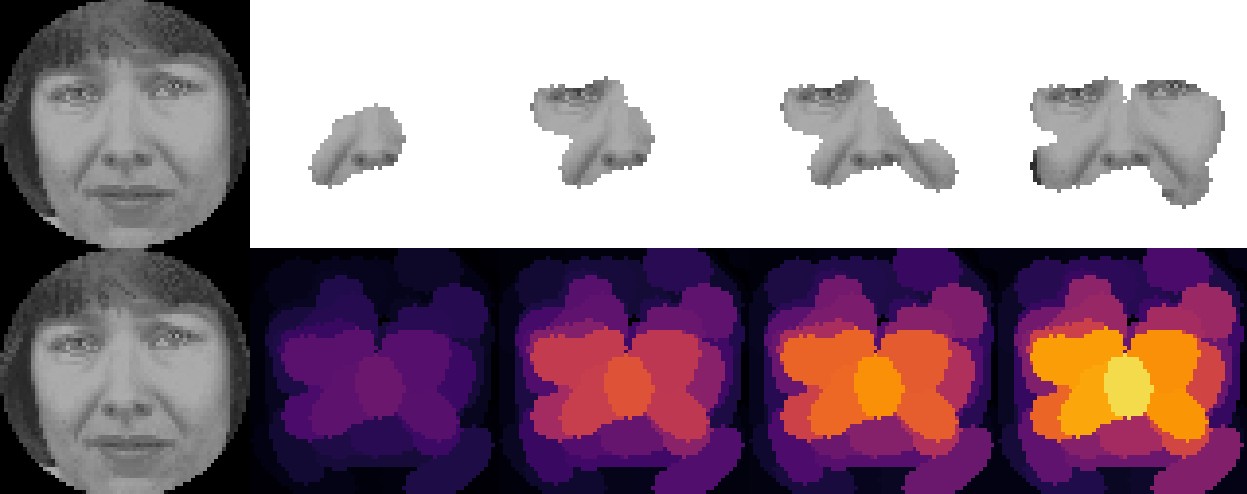

**Figure 10.** XRAI visualization of a man performing emotion "FEAR" that was classified as "HAPPINESS". Explanations corresponding to the KDEF image with id AF18SAS.

We also found that only 30–40% of the photographs taken with half profile angles were correctly classified, while the straight angle has a positive accuracy rate of 53%. This suggests that the model has some trouble identifying the relevant characteristics when the face of the subject is turned, even if it detects them on the straight angle. To check this, we can use the explanations that XRAI provided, comparing the important regions of two pictures of the same subject but from different angles. Figure 11 shows a woman with an angry pose from two angles, and is a representative example of these cases.

As we can see in Figure 12, which shows the XRAI explanation of the straight angle's prediction (Figure 11, left), the model has no problem in recognizing the woman frowning and focuses both on her eyebrows and her mouth. Meanwhile, on the same pose, captured from a half left profile angle (Figure 11, right), the model cannot recognize these features and instead focuses on a strand of hair, as seen in Figure 13. This strand of hair does not even appear on the photographs taken of this woman while posing as fearful, which means that it must have learned this characteristic from another subject.

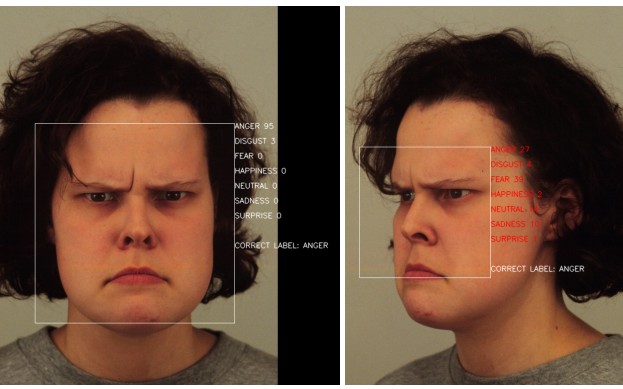

**Figure 11.** (**left**) Image classified as "ANGER" by the model. (**right**) Image classified as "FEAR" by the model. Predictions corresponding to the KDEF image with id AF16ANHL.

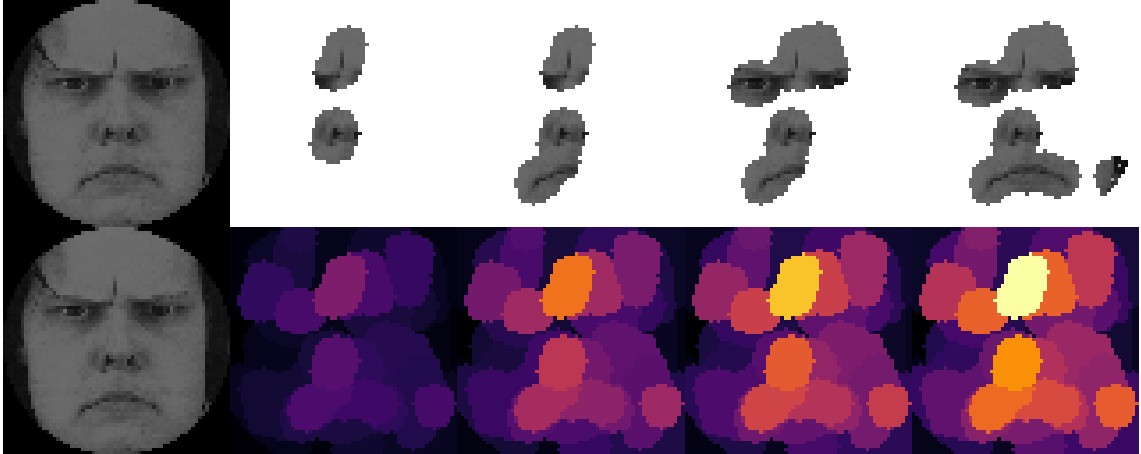

**Figure 12.** XRAI visualization of a woman performing emotion "ANGER" that was classified as "ANGER". Explanations corresponding to the KDEF image with id AF16ANS.

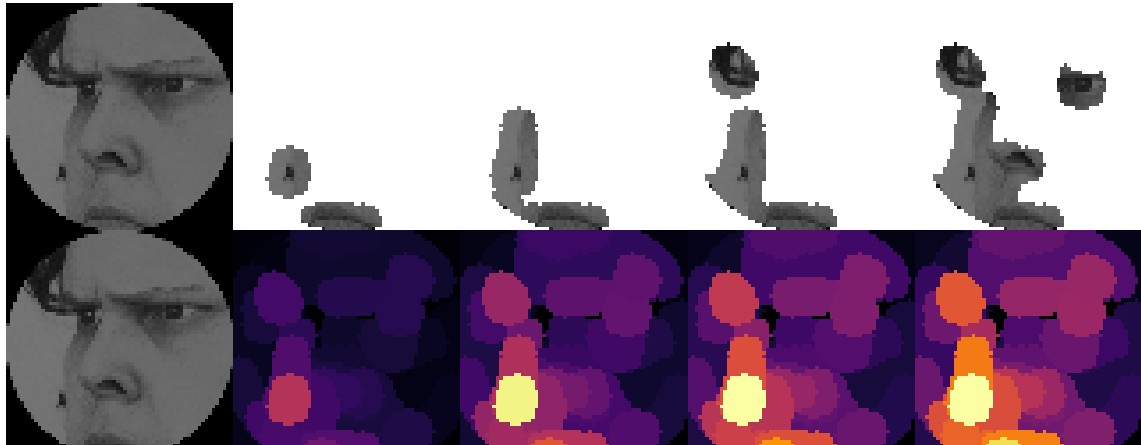

**Figure 13.** XRAI visualization of a woman performing emotion "ANGER" that was classified as "FEAR". Explanations corresponding to the KDEF image with id AF16ANHL.

After testing the model with *KDEF*, we used the *RaFD* dataset to test the model against unknown pictures. As with the previous dataset, we found that the model performs considerably worse on images taken from a half profile angle, with 75% mislabelled images, and works better with a straight angle, with 37% mislabelled images. Considering all angles, the model is able to classify correctly 530 of 1398 images (38%).

In this dataset, we found problems similar to those encountered while testing with *KDEF*. Those characteristics that the model has been able to learn are applied to these new photographs, while many instances are classified based on unimportant attributes. Figure 14 shows and example of a correctly classified instance of this dataset, and Figure 15 shows a mislabeled one.

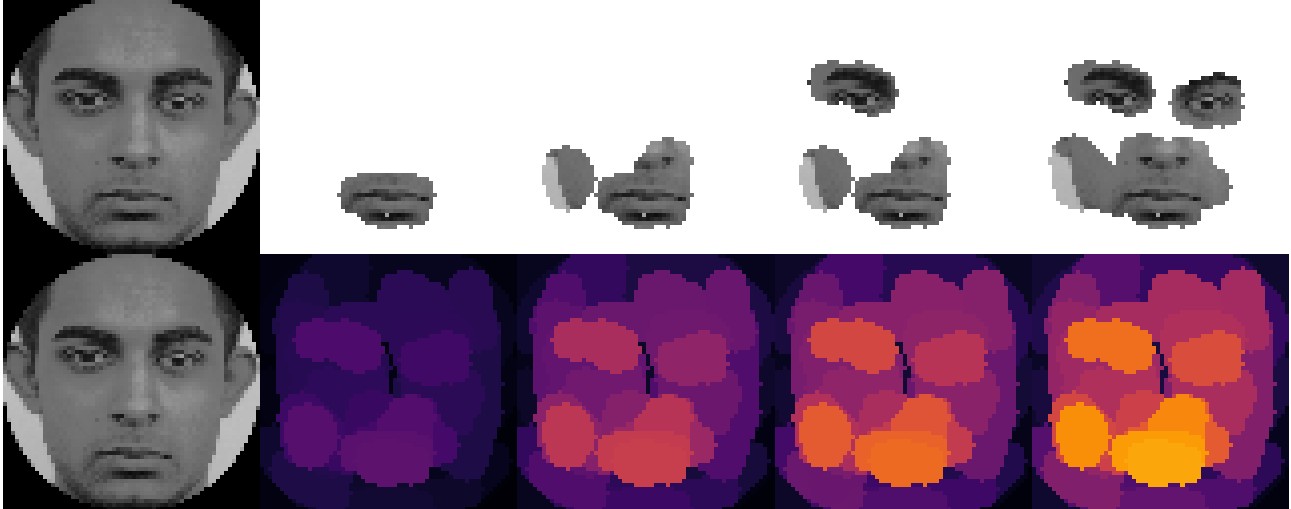

**Figure 14.** XRAI visualization of a man performing emotion "NEUTRAL" that was classified as "NEUTRAL". Explanations corresponding to the RaFD image with id 090_73.

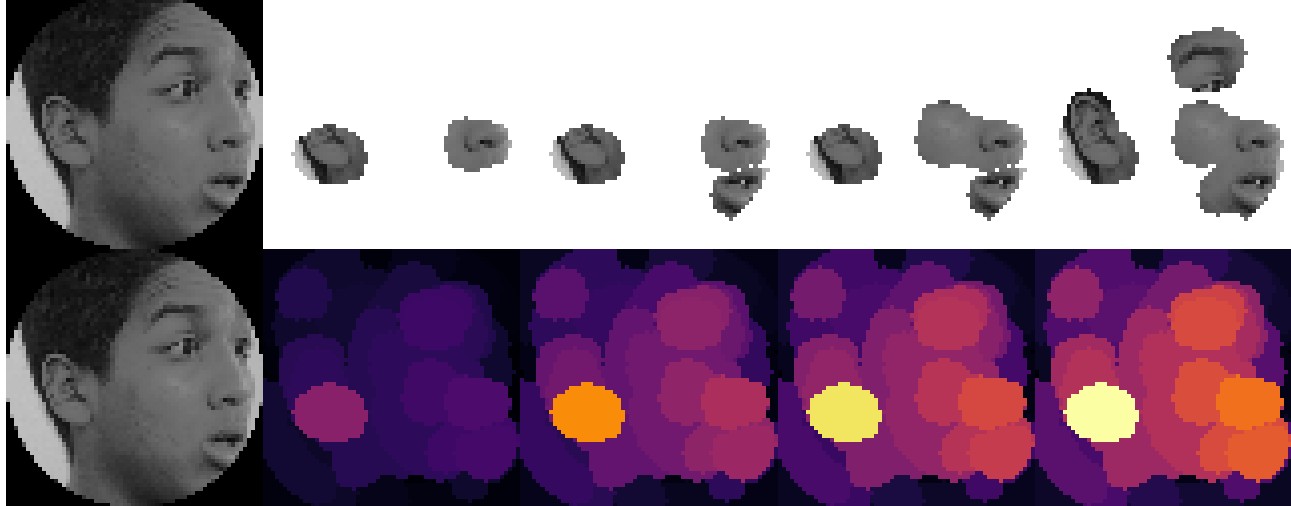

**Figure 15.** XRAI visualization of a man performing emotion "SURPRISE" that was classified as "NEUTRAL". Explanations corresponding to the RaFD image with id 045_67.

The most important lesson we learn from these explanations is that the model has not been able to correctly infer the characteristics of each emotion. A possible alternative to solve these problems would be to train it with more images to improve prediction. Since the model struggles to classify correctly photographs that were not taken from a straight angle, *RaFD* would be a great addition for training due to the multiple angles of its images, which could help the model recognize the important characteristics of each emotion from other angles.

### 4.2.2. Activity Model's Results

To test the activity model, we used *3MDAD* first. Because this dataset records each subject performing activities during multiple seconds, frame by frame, it contains more

images than necessary. We chose to use some of the frames for testing, specifically Frames 15, 30, 45, and 60 of each pair of subject and activity. This left us with 2000 pictures to which XRAI was applied.

In this case, only 60 images (3%) were incorrectly classified, which is reasonably accuracy for data used in testing. If we look at the mislabelled pictures, some of them are frames that do not represent the activity that the driver is supposed to be doing or that are ambiguous, so the given output could be considered a correct prediction.

Figure 16 illustrates an example of such "incorrectly" classified pictures. In the image, we can see an abbreviation of the prediction made by the model (e.g., "MESSAGE L" means "Sending a message with the left hand").

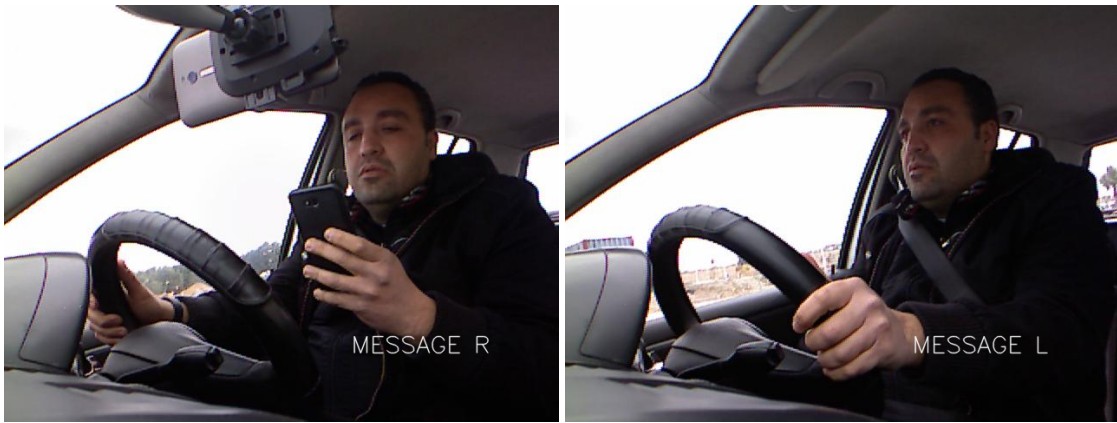

**Figure 16.** (**left**) Frame 15 of Subject 7 from 3MDAD performing activity "TALKING R" that was classified as "MESSAGE R". (**right**) Frame 15 of Subject 7 from 3MDAD performing activity "DRINKING L" that was classified as "MESSAGE L".

It is interesting to look at the explanations for these mislabelled examples to see if the model's logic is well founded. As we can see in Figure 17, the first prediction can be considered accurate, since the model locates the driver looking at his phone and supposes that the driver is sending a message.

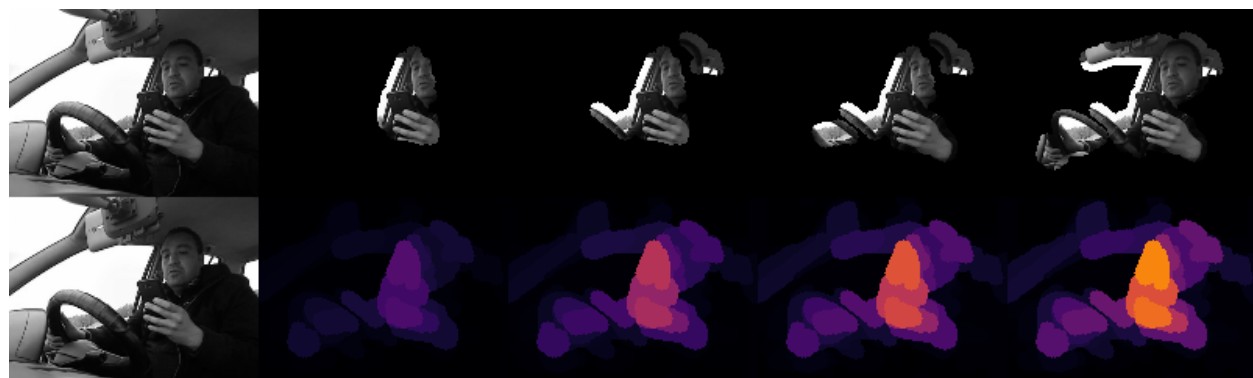

**Figure 17.** XRAI visualization of Frame 15 of Subject 7 from 3MDAD performing activity "TALKING R" that was classified as "MESSAGE R".

However, if we look at Figure 18, we can observe that the second mislabelling is not justified. By looking at Figure 16, one could think that, since the driver's left hand is not on the wheel, the model could interpret that the driver is using their phone instead of holding a cup. Yet, if we look at the XRAI explanation, we can see that the model does not focus on the absent hand but on the driver, only noticing the left side of the wheel within the top 20% of important regions.

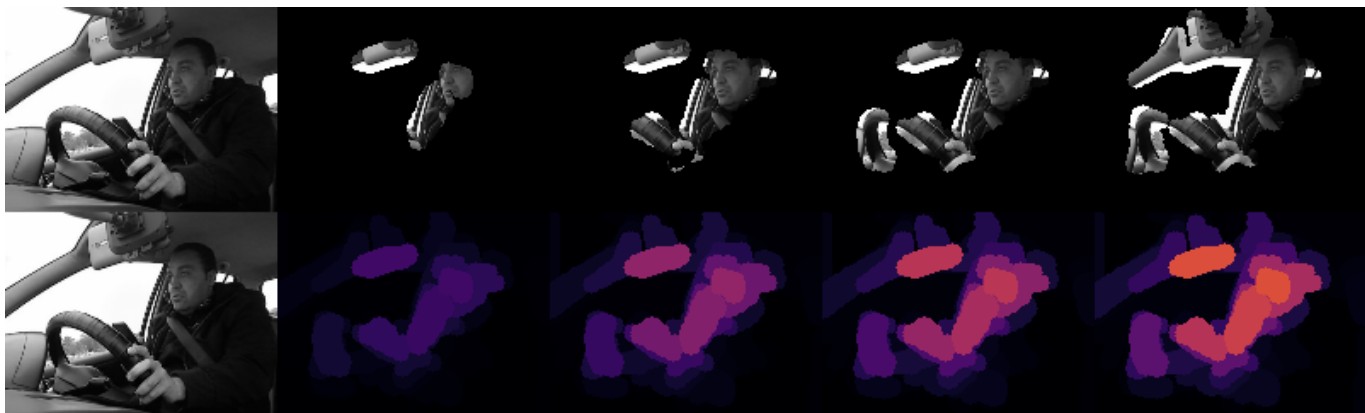

**Figure 18.** XRAI visualization of Frame 15 of Subject 7 from 3MDAD performing activity "DRINKING L" that was classified as "MESSAGE L".

If we look at the explanations of the correctly classified instances of activity "DRINK-ING L", we can deduce the reasons behind this mislabelling. As we can observe in Figure 19, because of the posture that the driver adopts while texting with his left hand, the phone is poorly visible. This means that the model cannot learn to locate the phone, and even when it classifies the frame correctly, it focuses on the hand of the driver and his posture instead of the phone, as we can see in Figure 20.

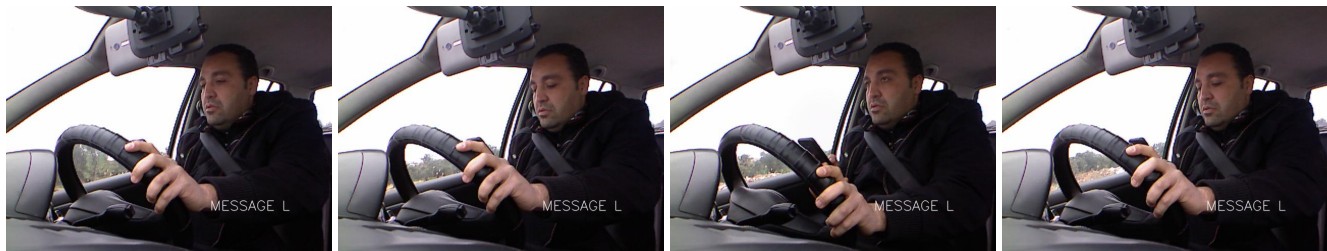

**Figure 19.** Frames 15, 30, 45, and 60 of Subject 7 from 3MDAD performing activity "MESSAGE L".

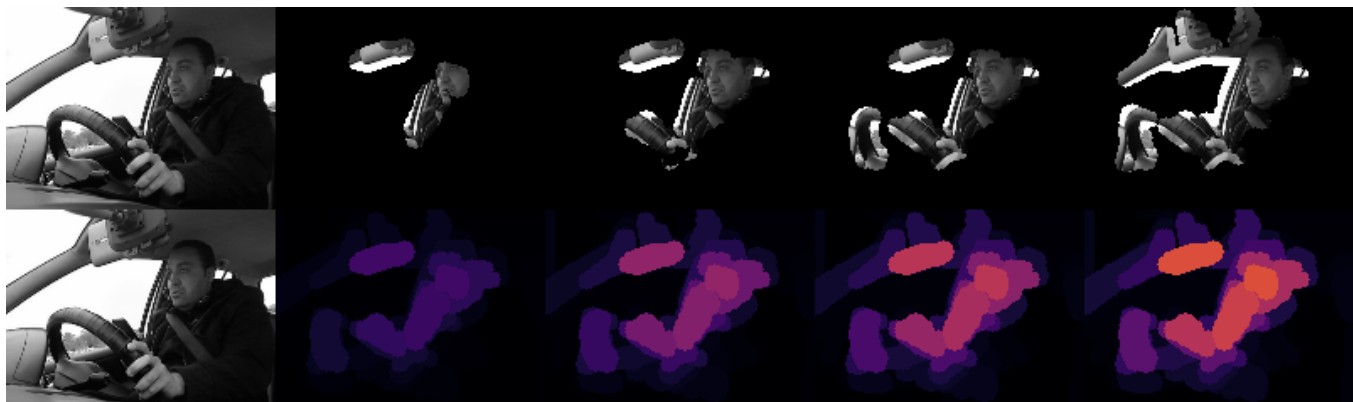

**Figure 20.** XRAI visualization of Frame 45 of Subject 7 from 3MDAD performing activity "TEXTING L" that was classified as "TEXTING L".

We find more of these cases in which the model classifies instances correctly by learning the wrong characteristics of the image. One clear example is Subject 27 performing "checking GPS" activity, as seen in Figure 21.

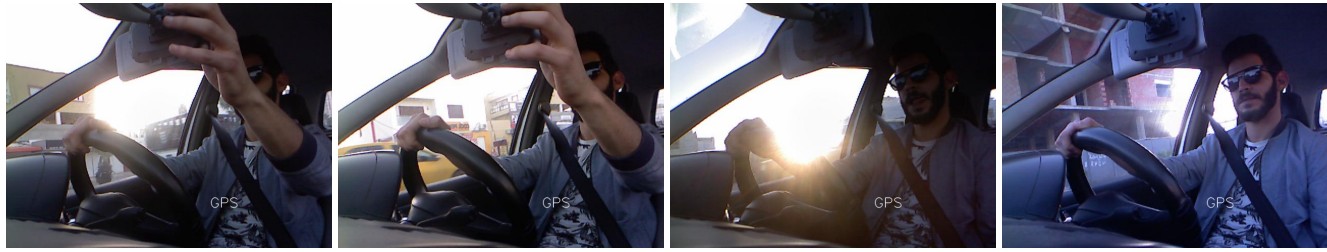

**Figure 21.** Frames 15, 30, 45, and 60 of Subject 27 from 3MDAD performing activity "GPS".

As we can see, the 15th and 30th frames show the driver using his phone to check GPS, while the 45th and 60th frames show what we could consider safe driving. Nevertheless, those last two frames are also classified correctly as "Checking GPS" due to the specific environmental conditions observed in the image. Figure 22 shows the explanation for Frame 60 classification, where we can see that the model focuses on the combination of the light and the face of the driver, and even the building behind.

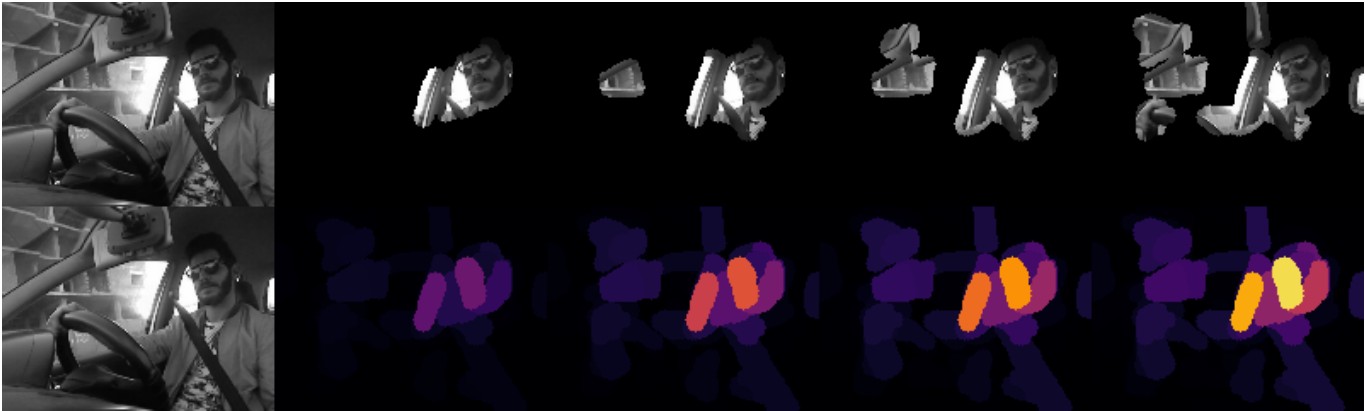

**Figure 22.** XRAI visualization of Frame 60 of Subject 27 from 3MDAD performing activity "GPS" that was classified as "GPS".

All of this proves that, to build an accurate model, one pair of subject and activity is not enough for each subject. It is necessary to record the same person doing the same activity in different poses, and with different clothes, illuminations, and backgrounds, so that the model does not learn the specific characteristics of the situation recorded. On the other hand, it is also important to train the model only with input data that is representative of its class.

However, XRAI proved to be an excellent method to validate the model. By analyzing the decisions taken, we can understand the scope and utility of the model. In this case, using only training data, we found that, even though accuracy was high, it is still necessary that the model learns to recognize generic elements instead of characteristics specific to some drivers.

Moving on to the other data, which we recreated by ourselves by imitating the *3MDAD* dataset, we expected the results to be worse, because this dataset was not used for training and it changed the subjects, the environment, and even the car. However, since the angle and position of the camera were similar, if the model was trained with enough data to generalize its deductions, then it should be able to classify correctly those new instances.

Figure 23 shows an example of a photograph recreated by us, next to an original one from 3MDAD. The GPS is absent in the photo, since we did not have the resources to attach it in all cars used for testing. This limited the number of photographs depicting the action "GPS" that we could take.

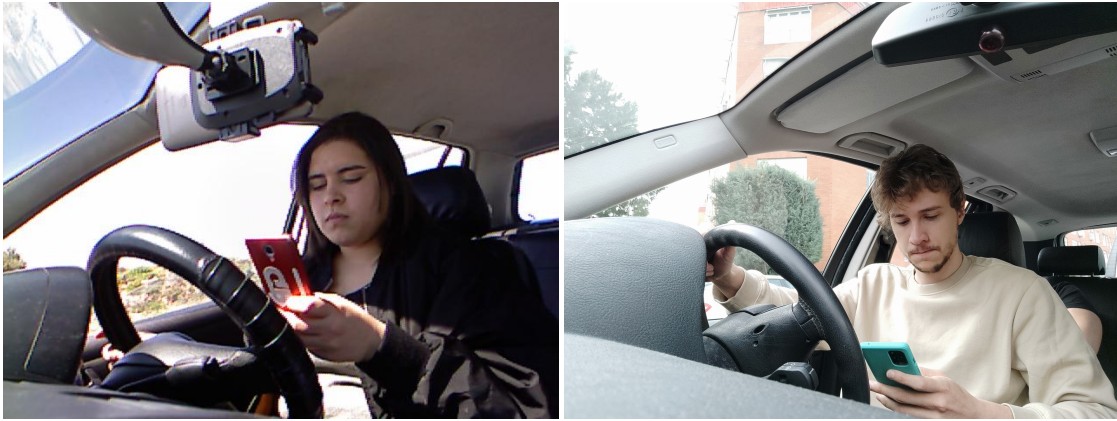

**Figure 23.** (**left**) Frame 60 of Subject 43 from 3MDAD performing activity "MESSAGE R". (**right**) Photograph taken by us of a subject performing activity "MESSAGE R".

After processing all 200 photographs, we found that only 78 of the images (39%) were correctly classified. This result by itself reveals that the model is not capable of generalization by using only the 3MDAD dataset at training, and this becomes even more evident when one looks at the misclassifications and realizes that most of the false positives are of instances classified as REACH BEH (39 out of 122, i.e., 31%), which shows that something is not working correctly. Following the methodology presented in this paper, we use the XRAI visualizations to understand why the model underperforms with the new data.

We start by searching for an explanation as to why the model gives so many false positives of the label "REACH BEH". To do this, we first examine two photographs: one that is correctly classified and one that it is not. These two examples can be seen in Figure 24.

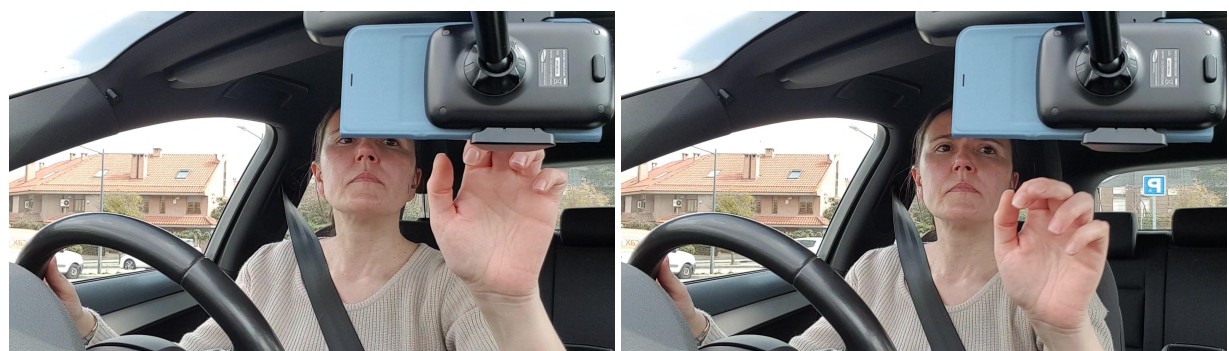

**Figure 24.** (**left**) Woman performing activity "GPS" that was classified as "GPS". (**right**) Woman performing activity "GPS" that was classified as "REACH BEH".

As we can see, the main difference between the two images is that in the second example the hand of the woman is sightly lowered, as is her head. This simple difference makes the model stop focusing on the hand and instead tries to classify the photograph based mostly on the woman's head position and on the elements around her. Figures 25 and 26 show the corresponding XRAI explanations.

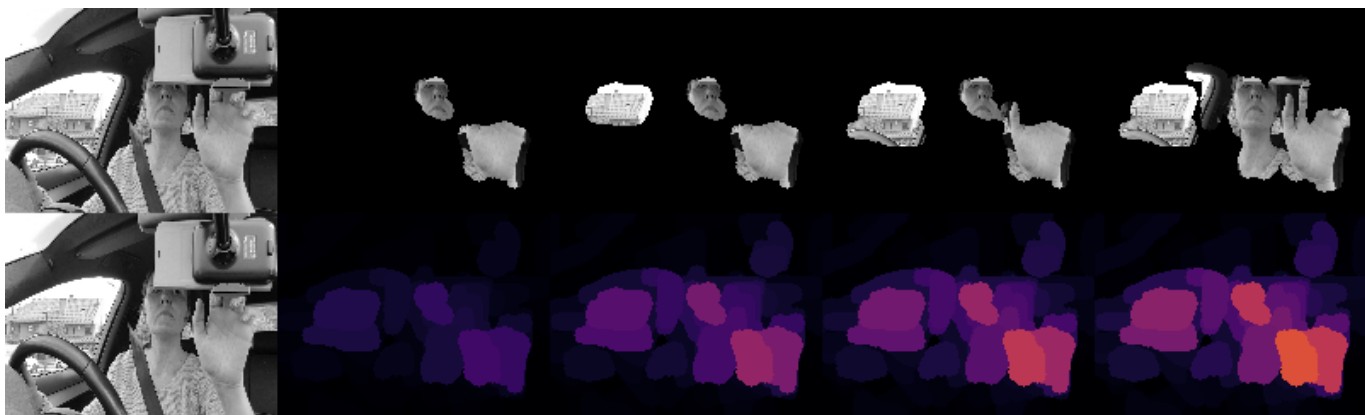

**Figure 25.** XRAI visualization of a woman performing activity "GPS" that was classified as "GPS".

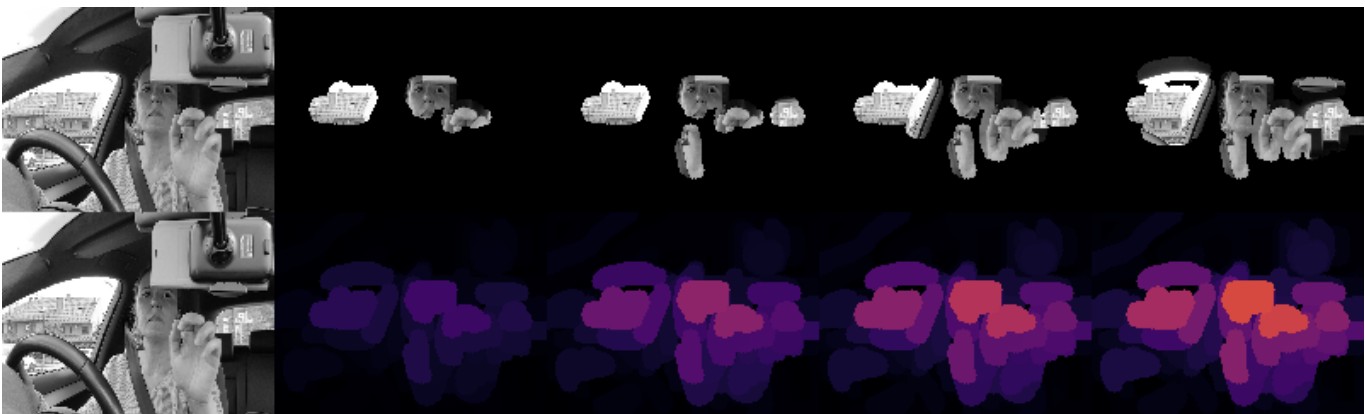

**Figure 26.** XRAI visualization of a woman performing activity "GPS"' that was classified as "REACH BEH".

While performing activity "REACH BEH", where the driver tries to reach something in the backseat, it is usual that he or she tries to keep their gaze on the road. Because of this, it would be reasonable that the model detects this action analyzing the driver's posture: one hand on the wheel, a turned head, body leaning to the side, etc. However, the model has not been able to learn how to exactly differentiate the "REACH BEH" posture from the rest, and it uses those characteristics even when there are other, more "obvious" elements in the photograph (in this case, it would be the hand close to the GPS, while in other cases it could be a visible bottle, having both hands on the wheel).

Another characteristic of the model that we can observe in these photos is that it uses information from across the window, possibly considering the scenery behind in its classification. This should be irrelevant if we want the model to work in any situation, since the driver could be anywhere. To avoid this, together with extending the training dataset so that it has more varied conditions, it could be interesting to explore a segmentation approach were we erase environment details, as in Xing et al.'s work [74].

Another common confusion among the labels is that the activities that involve holding a phone in some way (that is, sending a message or talking on the phone) are often confused with the action of drinking something. To analyze these situations, we will use the two examples presented in Figure 27.

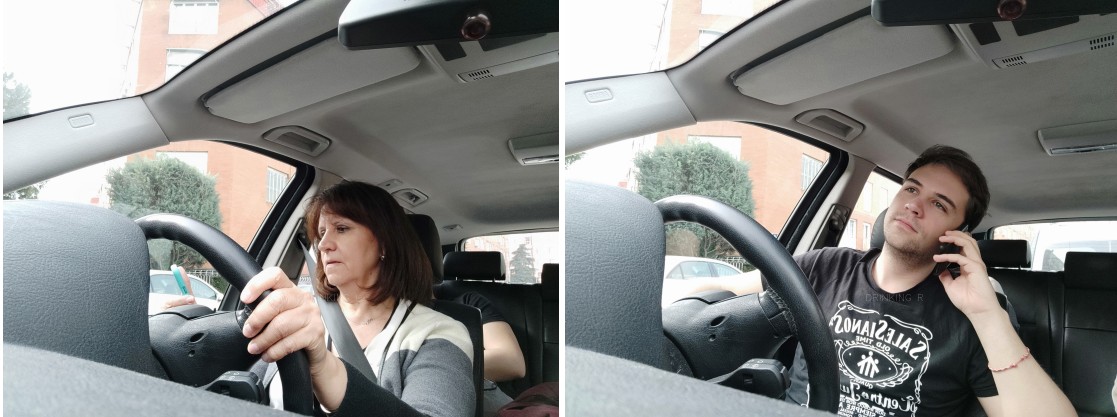

**Figure 27.** (**left**) Woman performing activity "MESSAGE L" that was classified as "DRINKING L". (**right**) Man performing activity "TALKING R" that was classified as "DRINKING R".

Figures 28 and 29 show the explanations that XRAI provided for the two situations in Figure 27. If we examine them, we can observe that the model is able to detect the hands and the general posture of the driver and that it notices that the driver is holding something. However, the model mistakes what the driver is holding and makes the incorrect assumption that it is a bottle or a cup, hence the "DRINKING" classification. To avoid these confusions, it could be useful to have an auxiliary model specialized in object recognition that could differentiate between phones, bottles, and cups, so that it could support the decision made by the activity model.

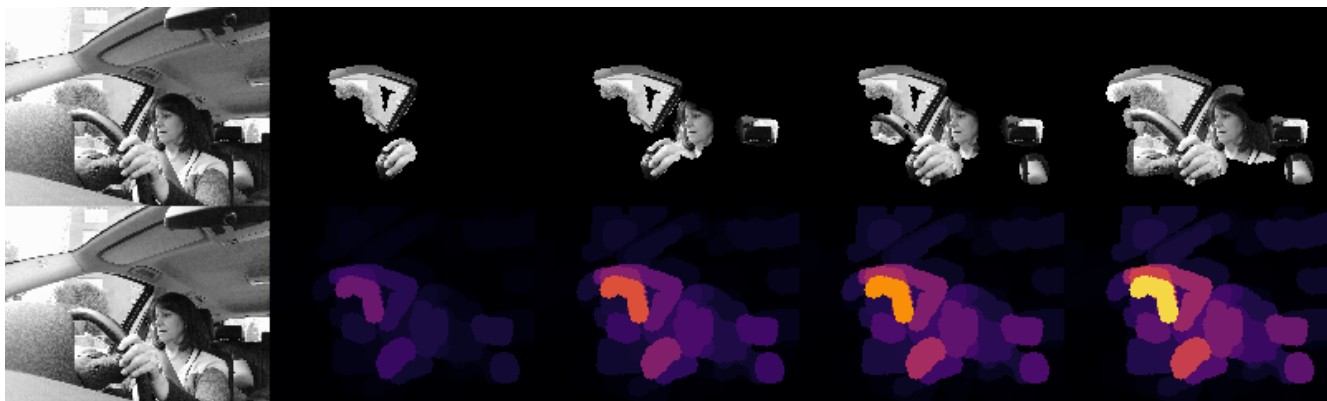

**Figure 28.** XRAI visualization of a woman performing activity "MESSAGE L" that was classified as "DRINKING L".

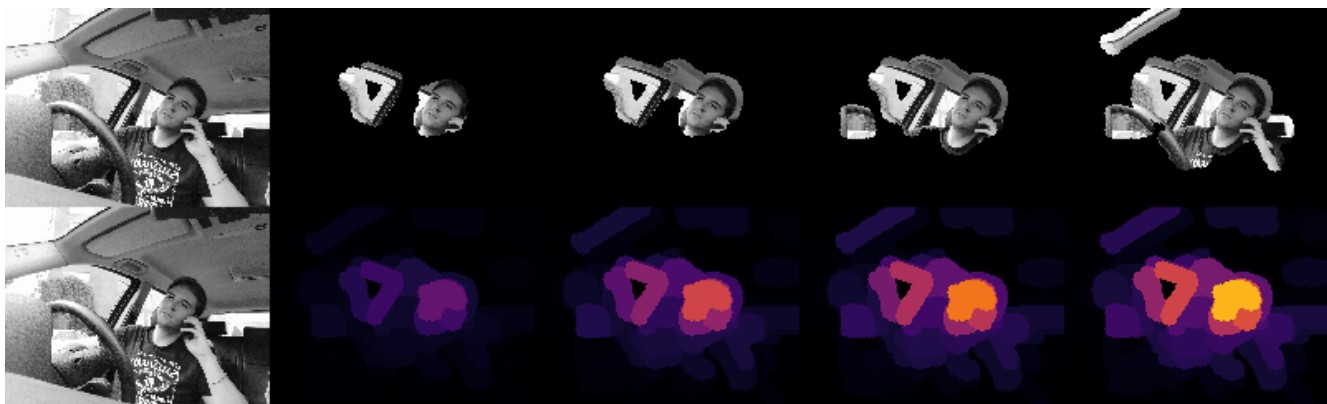

**Figure 29.** XRAI visualization of a man performing activity "TALKING R" that was classified as "DRINKING R".

However, it is also interesting to look at the examples that have been correctly classified. Among these images, we find interesting cases as the one displayed in Figure 30. Taking into account that the image is mirrored, as when taking a selfie with your phone, the image shows a man taking a bottle with his left hand, while his right hand stays on the steering wheel. Since the bottle is at the rightmost part of the image, it is not surprising that the model thought that he was drinking with his right hand. However, as Figure 31 shows, the model is capable of locating his right hand on the steering wheel, and it classifies the image correctly as "DRINKING L".

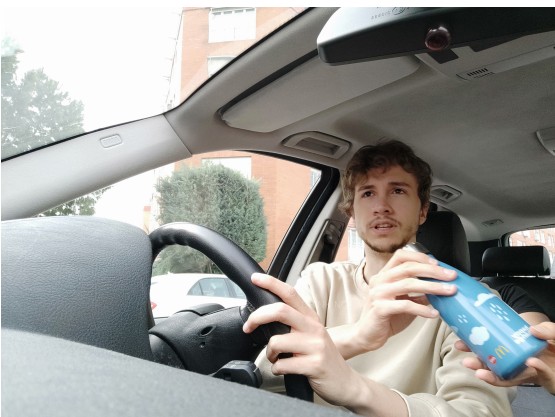

**Figure 30.** Man performing activity "DRINKING L" that was classified as "DRINKING L".

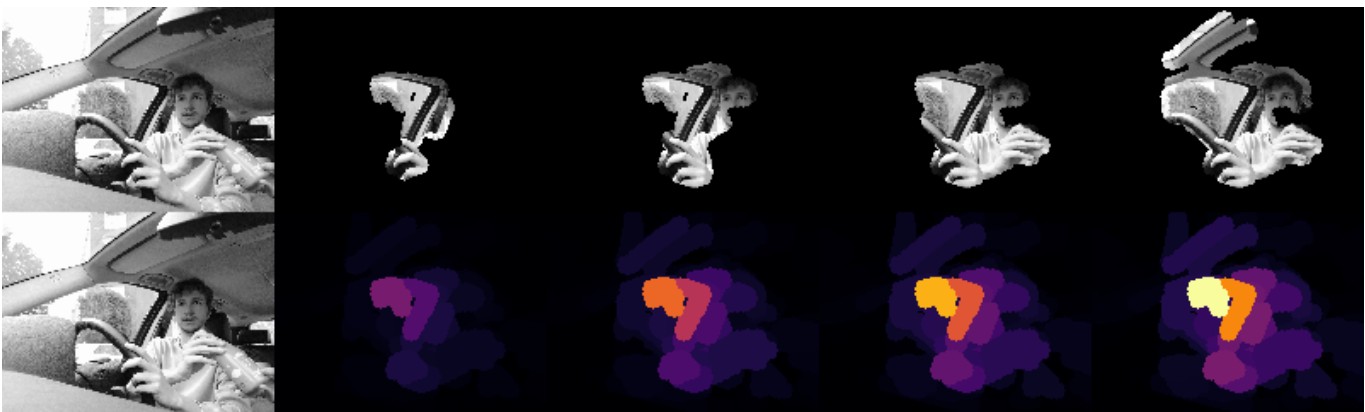

**Figure 31.** XRAI visualization of a man performing activity "DRINKING L" that was classified as "DRINKING L".

From this experimentation, we can conclude that, even though this model has potential and can recognize some interesting features of the activities performed, it still needs massive amounts of new data to be able to generalize and classify correctly new images. *3MDAD*, the dataset used for training, is a large database with more than 100,000 images, but does not have that much variety in terms of people, clothes, and general environment conditions.

The activity model presented in [7] was trained and tested using a subset of the images for validation and another one for test, and positive results were obtained. However, after seeing the model's performance on the data external to *3MDAD*, we can only conclude that the positive results were obtained due to the similarities between images. After all, even if the model is trained with, for example, 70% of the frames of all subjects, the remaining 30% of the frames are probably quite similar, because they are the same people and the illumination conditions are essentially the same. By training the model with more people and more situations, the model should be able to generalize better.

## 5. Conclusions and Future Work

In this work, we used *XRAI* [60], an explainable artificial intelligence technique, to understand how an ADAS works. We analyzed predictions provided by the two deep neural networks that form an ADAS [7], where one detects the mood of a driver and the other can detect if that driver is performing a distracting activity.

By using *XRAI*, we could observe that the main problem with the emotions model was the insufficient amount of training, because the model does not always focus on the main characteristics of each emotion (eyebrows, mouth, etc.) and fails as a result. This is specially remarkable when the picture is not taken from a straight angle, so the model would need to be trained with rich and varied amounts of data, taken from different angles.

On the other hand, although the activity model has a high accuracy on the training dataset, we could observe that it is not able to generalize too well and does not classify new data correctly. XRAI explanations gave us a better insight into how this activity model makes its decisions, which allowed us to identify possible improvement methods, such as using a segmentation approach or providing the ADAS with an object-recognition system that could help increase the accuracy of the model. Apart from this, the model would greatly benefit from more training, for which new and varied data would be needed.

This work proves the efficacy of post-hoc explainability techniques for interpreting and understanding the decisions taken by a black-box model such as deep neural networks. We used the *XRAI* technique to debug the models, but it can provide equally useful explanations to the driver so that he or she could understand the decisions taken by the ADAS.

**Author Contributions:** M.P.S.L.L., J.A.I.M. and A.S.d.M.; methodology, A.L.E.; software, E.M.L. and L.A.F.; validation, E.M.L.; formal analysis, M.P.S and J.A.I.M.; investigation, M.P.S.L. and A.S.d.M.; resources, A.L.E.; data curation, E.M.L. and L.A.F.; writing—original draft preparation, M.P.S.L., E.M.L. and J.A.I.M.; writing—review and editing, A.S.d.M. and A.L.E.; visualization, A.L.E. and J.A.I.M.; supervision, M.P.S.L. and A.S.d.M.; project administration, A.S.d.M.; funding acquisition, A.S.d.M. and A.L.E. All authors contributed to manuscript revision, read, and approved the submitted version.

**Funding:** This research received no external funding.

**Data Availability Statement:** Data not available.

**Acknowledgments:** This work was supported under projects PEAVAUTO-CM-UC3M, PID2019-104793RB-C31, and RTI2018-096036-B-C22, and by the Region of Madrid's Excellence Program (EPUC3M17).

**Conflicts of Interest:** The authors declare no conflict of interest.

## Appendix A. Extended Results: Confusion Matrices

In this section, we detail how many times the models were able to classify correctly the images of the datasets used by showing a confusion matrix for each dataset. Figures A1–A3 correspond to the results obtained from the classification of the datasets KDEF, RaFD, and 3MDAD, respectively. Figure A4, on the other hand, shows the activity model's performance over the photographs taken by us, thereby imitating the *3MDAD* dataset.

## Confusion matrix (KDEF's results)

| True label \ Predicted label | ANGER | DISGUST | FEAR | HAPPINESS | NEUTRAL | SADNESS | SURPRISE |
|---|---|---|---|---|---|---|---|
| ANGER | 376 | 17 | 27 | 41 | 35 | 95 | 16 |
| DISGUST | 89 | 184 | 28 | 22 | 36 | 138 | 15 |
| FEAR | 58 | 1 | 236 | 73 | 24 | 94 | 50 |
| HAPPINESS | 9 | 3 | 53 | 530 | 18 | 54 | 18 |
| NEUTRAL | 68 | 1 | 19 | 52 | 238 | 128 | 32 |
| SADNESS | 46 | 5 | 39 | 62 | 36 | 426 | 18 |
| SURPRISE | 41 | 1 | 81 | 113 | 26 | 47 | 210 |

**Figure A1.** Results obtained by the emotion recognition model over the KDEF dataset.

## Confusion matrix (RaFD's results)

| True label \ Predicted label | ANGER | DISGUST | FEAR | HAPPINESS | NEUTRAL | SADNESS | SURPRISE |
|---|---|---|---|---|---|---|---|
| ANGER | 246 | 1 | 13 | 7 | 15 | 31 | 11 |
| DISGUST | 81 | 78 | 17 | 6 | 10 | 31 | 16 |
| FEAR | 18 | 1 | 90 | 14 | 43 | 22 | 56 |
| HAPPINESS | 15 | 1 | 5 | 230 | 29 | 20 | 16 |
| NEUTRAL | 35 | 0 | 9 | 6 | 136 | 57 | 25 |
| SADNESS | 48 | 2 | 20 | 15 | 23 | 128 | 28 |
| SURPRISE | 16 | 0 | 30 | 38 | 17 | 20 | 152 |

**Figure A2.** Results obtained by the emotion recognition model over the RaFD dataset.

## Confusion matrix (3MDAD's results)

| True label \ Predicted label | SAFE DRIVING | FATIGUE | DRINKING R | DRINKING L | REACH BEH | GPS | MESSAGE R | MESSAGE L | TALKING R | TALKING L |
|---|---|---|---|---|---|---|---|---|---|---|
| SAFE DRIVING | 396 | 0 | 0 | 0 | 0 | 2 | 0 | 0 | 0 | 0 |
| FATIGUE | 1 | 384 | 3 | 0 | 3 | 0 | 0 | 0 | 0 | 1 |
| DRINKING R | 0 | 1 | 396 | 0 | 2 | 1 | 2 | 0 | 0 | 0 |
| DRINKING L | 2 | 0 | 0 | 388 | 1 | 0 | 0 | 2 | 0 | 0 |
| REACH BEH | 0 | 1 | 1 | 1 | 380 | 1 | 0 | 3 | 0 | 0 |
| GPS | 4 | 1 | 0 | 1 | 2 | 326 | 1 | 0 | 0 | 0 |
| MESSAGE R | 0 | 0 | 0 | 0 | 0 | 0 | 398 | 0 | 1 | 0 |
| MESSAGE L | 0 | 1 | 1 | 2 | 1 | 0 | 1 | 388 | 0 | 0 |
| TALKING R | 0 | 0 | 0 | 0 | 0 | 3 | 3 | 0 | 388 | 0 |
| TALKING L | 3 | 2 | 0 | 0 | 0 | 0 | 0 | 5 | 0 | 380 |

**Figure A3.** Results obtained by the activity recognition model over the 3MDAD dataset.

**Confusion matrix (Own photographs' results)**

| True label \ Predicted label | SAFE DRIVING | FATIGUE | DRINKING R | DRINKING L | REACH BEH | GPS | MESSAGE R | MESSAGE L | TALKING R | TALKING L |
|---|---|---|---|---|---|---|---|---|---|---|
| SAFE DRIVING | 0 | 4 | 1 | 2 | 5 | 1 | 0 | 1 | 2 | 1 |
| FATIGUE | 0 | 2 | 2 | 1 | 3 | 0 | 0 | 0 | 1 | 0 |
| DRINKING R | 0 | 0 | 11 | 4 | 4 | 2 | 5 | 0 | 0 | 0 |
| DRINKING L | 0 | 2 | 0 | 17 | 4 | 0 | 2 | 0 | 0 | 0 |
| REACH BEH | 0 | 0 | 1 | 1 | 18 | 1 | 0 | 0 | 0 | 1 |
| GPS | 0 | 1 | 2 | 1 | 2 | 3 | 0 | 0 | 0 | 0 |
| MESSAGE R | 1 | 0 | 4 | 0 | 5 | 2 | 8 | 0 | 4 | 1 |
| MESSAGE L | 0 | 2 | 3 | 3 | 5 | 0 | 2 | 1 | 0 | 0 |
| TALKING R | 0 | 0 | 4 | 1 | 7 | 0 | 0 | 0 | 15 | 1 |
| TALKING L | 0 | 1 | 2 | 5 | 4 | 1 | 4 | 1 | 0 | 3 |

**Figure A4.** Results obtained by the activity recognition model over the new photographs taken by us.

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
