# Peer review of "Explaining Deep Learning-Based Driver Models"

_applsci, doi:10.3390/app11083321_

Round 1
Reviewer 1 Report
The core of the paper's contribution is related to the utilization of "Explainable Artificial Intelligence" in order to understand the inner functional logic of an advanced driver assistance system. The contribution is interesting, but it should be optimized considering some relevant aspects.
1. The whole investigative approach should be better linked to the real-world scenarios, so that any reader can perfectly understand how the logical core of any ADAS system may benefit from the consideration of the results that are reported.
2. The authors claim that, apart from explaining the inner logical core, they also bring some improvements to the system. The improvements that were made should be made clearer in the paper's text.
3. I would suggest that the authors describe the test infrastructure that was used, including the hardware and software features.
4. The English language should be fully proofread and improved.
Author Response
Paper Applied Science: applsci-1144838
Explaining deep learning-based driver models
We are thankful to the reviewer 1 and the Editor for their useful comments and constructive criticism. The changes we made in the revised revision in response to the respective referees are detailed as follows:
Reviewer 1
The core of the paper's contribution is related to the utilization of "Explainable Artificial Intelligence" in order to understand the inner functional logic of an advanced driver assistance system. The contribution is interesting, but it should be optimized considering some relevant aspects.
Question 1
The whole investigative approach should be better linked to the real-world scenarios, so that any reader can perfectly understand how the logical core of any ADAS system may benefit from the consideration of the results that are reported.
Answer 1
Thank you for the comment, it is indeed useful four our research, this aspect has been considered in the revised manuscript. The introduction has been modified in order to clarify how our proposal is linked to the real-world scenario. A figure (Figure 1) has been included so it can be better understood the contribution of the research proposed.
Question 2
The authors claim that, apart from explaining the inner logical core, they also bring some improvements to the system. The improvements that were made should be made clearer in the paper's text.
Answer 2
We must clarify that the system was not modified in any way, we just brought up some possible improvements that could be done to the system. We have modified the text to avoid sentences that could be misleading in this aspect:
“In this work, a model of driver emotion detection and a model of driver activity detection will be analyzed using XAI techniques in order to explain how these models work. The explanation of these models, which are an important part of an ADAS, can serve many different purposes such as debugging them and identifying possible improvements. More precisely, the two different models considered in this research are: a model that has been trained to detect the driver’s mood, and a model that can detect whether the driver is attentive or distracted (for example, texting on the phone).”
Question 3
I would suggest that the authors describe the test infrastructure that was used, including the hardware and software features.
Answer 3
In the revised manuscript, we have specified the whole test infrastructure in section 4.2 “Results and discussion”:
“The experimentation was performed by using a Python program that could be executed in virtually any computer. This program loads the models, reads a batch of images, processes them and then provides the XRAI explanation of the predictions. The most relevant aspects of the testing environment are the libraries and packages used, which would be Tensorflow 2.3.1, Keras 2.4.3, Pyhton 3.8.5 and OpenCV 4.4.0.46. We also used the XRAI implementation provided in Saliency package, version 0.0.5.”
Question 4
The English language should be fully proofread and improved.
Answer 4
The revised manuscript has been proofread and improved as suggested. In addition, the final version of the paper could be revised by the MDPI's English editing service, if needed.

Reviewer 2 Report
The paper deals with using an Explainable Artificial Intelligence to recognize driver’s behavior for future ADAS improvement. The introduction is slightly wordy, but the state of art XAI techniques are detailed and succeed in explaining the choice made by the Authors. The chapter about emotion recognition is unnecessarily long and not too useful for the paper's final aim.
As stated by the Authors, the XRAI technique has been trained using two datasets: one dedicated to the emotions and the activity. The results are useful for images that are similar to the one belonging to the training dataset but at the same time are not satisfactory for pictures taken at different angles or in a different environment.
The paper provides an essential basis for the use of this XAI technique in the field of ADAS, but the idea needs to be further developed in order to make the method useful in different types of vehicles. In this context, the conclusions of the Authors are right and they point towards the right direction.
Author Response
Paper Applied Science: applsci-1144838
Explaining deep learning-based driver models
We are thankful to the reviewer 2 and the Editor for their useful comments and constructive criticism. The changes we made in the revised revision in response to the respective referees are detailed as follows:
Reviewer 2
The paper deals with using an Explainable Artificial Intelligence to recognize driver’s behavior for future ADAS improvement.
Question 1
The introduction is slightly wordy, but the state of art XAI techniques are detailed and succeed in explaining the choice made by the Authors. The chapter about emotion recognition is unnecessarily long and not too useful for the paper's final aim.
Answer 1
The revised manuscript has been changed as suggested. We have modified the introduction with a better idea about the goal of the paper and the chapter about emotion recognition has been reduced as suggested.
Question 2
As stated by the Authors, the XRAI technique has been trained using two datasets: one dedicated to the emotions and the activity. The results are useful for images that are similar to the one belonging to the training dataset but at the same time are not satisfactory for pictures taken at different angles or in a different environment.
Answer 2
To avoid the inconsistency between databases, we have recreated photographs that are more similar to the images used in training for the model that recognizes activities. We think that the conclusions are now much more interesting and valuable.
Question 3
The paper provides an essential basis for the use of this XAI technique in the field of ADAS, but the idea needs to be further developed in order to make the method useful in different types of vehicles. In this context, the conclusions of the Authors are right and they point towards the right direction.
Answer 3
Thanks for your comment. The introduction has been modified in order to clarify how our proposal is linked to the real-world scenario. A figure (Figure 1) has been included so it can be better understood the contribution of the research proposed. We think that this aspect clarifies how our proposal is useful in different types of vehicles.

Reviewer 3 Report
The document reports a research work consisting on the application of a post-hoc explainable artificial intelligence (XAI) model to the problem of advanced driver-assistance systems (ADAS). The work is well motivated and well situated in relation to the state of the art. However some options taken by the authors in the experiments are questionable and consequently lead to very poor results. Please notice that I support publication of research with negative results, in case the methodology is sound and the experiments fail, allowing to disprove some research idea. Unfortunately that is not the case in this work.
The foremost reason for the above opinion is the mismatch of the training and testing conditions in the experiment of detecting activities while driving. The training set was composed of frontal images of the driver while the test set was composed of side images. To consider that the challenge was found interesting, as the authors do, is just puzzling. In a difficult classification problem as this one, to significantly change the conditions between the training and the testing phases is prone to lead to very poor results, preventing the extraction of meaningful conclusions.
The results of the emotion detection are also poor, although in this case there is no inconsistency between training and testing sets. Since the XAI used is of the post-hoc type, it would be interesting to know the result of the ADAS system in each of the cases. This could provide clues on what is failing the XAI.
This work is based on a ADAS system that was developed by one of the authors, but the reference [6] is incomplete and has a non-English title, which limits the reader.
Related to this, the bibliography needs a thorough revision. Incomplete references abound, including without title [6, 42, 48, 51, 67].
My suggestion is that authors redo the experiment with compatible data sets and also situate the results of the XAI model in comparison to the underlying ADAS results.
English needs to be revised.
Next I point out some detailed improvements that should be made.
- The title should replace “driver models” by “driver-assistance systems”
- line 66 “emotions of the drive” → “emotions of the driver”
- lines 124-125 “cumulative prospect theory (CPT)”, “inverse reinforcement learning”, and “nonlinear logistic regression” should all have references.
- line 219 “one of this ten common activities” → “one of these ten common activities”
- sec. 4.1.1 – the title of the section is “Training datasets” therefore the descriptions of the KDEF and 3MDAD datasets should read “train” and not “test”
- line 300 “on time” → “in time”
- in all figures (from 6 onwards) with XAI regions of interes, legends should indicate what the explanation pictures mean (possibly different percentages of top results…)
- line 335 consider “Said strand” → “The said strand”
Author Response
Paper Applied Science: applsci-1144838
Explaining deep learning-based driver models
We are thankful to the reviewer 3 and the Editor for their useful comments and constructive criticism. The changes we made in the revised revision in response to the respective referees are detailed as follows:
Reviewer 3
The document reports a research work consisting on the application of a post-hoc explainable artificial intelligence (XAI) model to the problem of advanced driver-assistance systems (ADAS).
Question 1
The work is well motivated and well situated in relation to the state of the art. However some options taken by the authors in the experiments are questionable and consequently lead to very poor results. Please notice that I support publication of research with negative results, in case the methodology is sound and the experiments fail, allowing to disprove some research idea. Unfortunately that is not the case in this work.
The foremost reason for the above opinion is the mismatch of the training and testing conditions in the experiment of detecting activities while driving. The training set was composed of frontal images of the driver while the test set was composed of side images. To consider that the challenge was found interesting, as the authors do, is just puzzling. In a difficult classification problem as this one, to significantly change the conditions between the training and the testing phases is prone to lead to very poor results, preventing the extraction of meaningful conclusions.
Answer 1
Thanks for your comment, you’re right, we focused too much on the fact that there weren’t similar databases available and we used a different one, and, while we found interesting the fact that it could recognize some things, it is true that it isn’t really useful for this paper.
We have followed your advice, and we have recreated photographs that are more similar to the images used in training. We think that the conclusions are now much more interesting and valuable, since we could detect precise improvements for the activity model thanks to the new images tested (we detected excessive false positives of one activity, and we also observed that the model tends to confuse the activities that involve bottles and phones).
Question 2
The results of the emotion detection are also poor, although in this case there is no inconsistency between training and testing sets. Since the XAI used is of the post-hoc type, it would be interesting to know the result of the ADAS system in each of the cases. This could provide clues on what is failing the XAI.
Answer 2
While we were writing the paper, we considered the option of detailing the underlying “numeric” ADAS results, but we decided to avoid giving them relevance in favor of the XRAI data, which is why we only provided the general accuracy percentages. However, it is true that providing more knowledge about the ADAS’ performance could help understand the XRAI results, so we have added the confusion matrices of the datasets tested as an Appendix.
Question 3
This work is based on a ADAS system that was developed by one of the authors, but the reference [6] is incomplete and has a non-English title, which limits the reader.
Answer 3
We have revised the reference and it is now included with an English title: “Driver emotion and behaviour recognition using Deep Learning”.
Question 4
Related to this, the bibliography needs a thorough revision. Incomplete references abound, including without title [6, 42, 48, 51, 67].
Answer 4
These references have been revised and updated.
Question 5
My suggestion is that authors redo the experiment with compatible data sets and also situate the results of the XAI model in comparison to the underlying ADAS results
Answer 5
We have completed the manuscript thanks to your reviews: we redid the experiments for the activity model where the angle was different than in the images used for testing, and we added the numeric results as an appendix, which we think complete the conclusions obtained.
Question 6
English needs to be revised.
Question 6
The revised manuscript has been proofread and improved as suggested. In addition, the final version of the paper could be revised by the MDPI's English editing service, if needed.
Question 7
Next I point out some detailed improvements that should be made.
- The title should replace “driver models” by “driver-assistance systems”
Answer 7
Thanks for your suggestions. However, in this research, as we have detailed in the introduction of the revised manuscript (Figure 1), driver modelling is only a part of the developed ADAS and therefore the concept of “Driver models” defines better our research than “Driver Assistance Systems”.
Question 8
- lines 124-125 “cumulative prospect theory (CPT)”, “inverse reinforcement learning”, and “nonlinear logistic regression” should all have references.
Answer 8
Thanks for your comments. All the references have been included in the revised manuscript (references [49, 50, 51]).
Question 9
- sec. 4.1.1 – the title of the section is “Training datasets” therefore the descriptions of the KDEF and 3MDAD datasets should read “train” and not “test”
Answer 9
Perhaps we didn’t make clear that those datasets are indeed being used for testing, with the main difference with the datasets presented in 4.1.2 being that these datasets were also used for training the model. We have changed the title of the section to “Datasets previously used for training” in hopes of it being more clear this way.
Question 10
- in all figures (from 6 onwards) with XAI regions of interest, legends should indicate what the explanation pictures mean (possibly different percentages of top results…)
Answer 10
All of the explanations are presented using the same percentages, which are explained in section 3.2 (just over the Figure 5 example). We considered that including the same percentages over and over would be counterproductive.
Question 11
Minor corrections
- line 66 “emotions of the drive” → “emotions of the driver”
- line 219 “one of this ten common activities” → “one of these ten common activities”
- line 300 “on time” → “in time”
- line 335 consider “Said strand” → “The said strand”
Answer 11
Thanks for your suggestions. All the corrections have been made in the revised manuscript.

Round 2
Reviewer 3 Report
The revision of the paper solved the major problem of inadequacy of the test data and several other aspects.
The new dataset of images produced by the authors could be very useful for other researchers, to reproduce this work or for new research work. Therefore I suggest authors to consider making it available with the paper, should it be published.
In line 314, a phrase indicating that the parameters used for fig. 6 are replicated in the following similar figures. It would help the reader.
Related to the previous, "multiple intensities" in legend of fig. 6 is not very informative. Is it just a variation of colour temperature for visual effect? Why isn't a single one enough (instead of 4)?